# Ask Language Model to Clean Your Noisy Translation Data

**Quinten Bolding**[⚒][*]    **Baohao Liao**[⚒]
**Brandon James Denis**[♟]    **Jun Luo**[♟]    **Christof Monz**[⚒]
[⚒]Language Technology Lab, University of Amsterdam
[♟]Huawei Amsterdam Research Center
quinten.bolding@gmail.com    b.liao@uva.nl

## Abstract

Transformer models have demonstrated remarkable performance in neural machine translation (NMT). However, their vulnerability to noisy input poses a significant challenge in practical implementation, where generating clean output from noisy input is crucial. The MTNT dataset (Michel and Neubig, 2018) is widely used as a benchmark for evaluating the robustness of NMT models against noisy input. Nevertheless, its utility is limited due to the presence of noise in both the source and target sentences. To address this limitation, we focus on cleaning the noise from the target sentences in MTNT, making it more suitable as a benchmark for noise evaluation. Leveraging the capabilities of large language models (LLMs), we observe their impressive abilities in noise removal. For example, they can remove emojis while considering their semantic meaning. Additionally, we show that LLM can effectively rephrase slang, jargon, and profanities. The resulting datasets, called C-MTNT, exhibit significantly less noise in the target sentences while preserving the semantic integrity of the original sentences. Our human and GPT-4 evaluations also lead to a consistent conclusion that LLM performs well on this task. Lastly, experiments on C-MTNT showcased its effectiveness in evaluating the robustness of NMT models, highlighting the potential of advanced language models for data cleaning and emphasizing C-MTNT as a valuable resource.[1]

## 1 Introduction

Neural machine translation (NMT) has witnessed significant progress (Bapna et al., 2018; Hieber et al., 2020; Liao et al., 2021) in recent years, particularly with the introduction of Transformer (Vaswani et al., 2017). Despite their impressive performance on clean benchmarks (Barrault et al., 2019; Huang et al., 2020; Liao et al., 2020), these models exhibit a noticeable decline in translation quality when exposed to noisy input. This hampers their performance in real-world scenarios, where human users unintentionally introduce misspellings and grammatical errors during text input (Karpukhin et al., 2019). Therefore, evaluating the robustness of NMT models against noisy inputs becomes crucial before their deployment.

Despite the importance of assessing the resilience of NMT models against noise, the available evaluation datasets remain limited. To the best of our knowledge, MTNT (Michel and Neubig, 2018) stands as one of the few well-established resources for evaluating NMT models' performance in the presence of noise. The noise distribution in MTNT closely resembles real-world use cases, but its applicability is constrained by the presence of noise in the target sentences. For instance, a French-English pair may appear as: "REEEEEEEEEEEE les normies sur mon eiffel y'en a marre" ↔ "REEEEEEEE bored of the normies on my eiffel", which is not desirable. Our expectation is that an effective NMT model is capable of translating a noisy source sentence into a clean target sentence.

In order to enhance the applicability of MTNT for evaluating NMT models, we propose cleaning the target side of this dataset. In contrast to conventional cleaning approaches (Xu and Koehn, 2017; Khayrallah et al., 2018; Koehn et al., 2020) that typically involve filtering out undesirable sentences and retaining only high-quality ones, we aim to remove noise from sentences without reducing the overall sample number. Language-aware rule-based approaches, which rely on predefined rules to eliminate noise, have been widely employed as a common method for such cleaning (Miao et al., 2021; Mavrogiorgos et al., 2022). While these methods can effectively remove certain types of noise, it becomes impractical to define rules for every possible noise source. Moreover, some natural noise introduced by human input is not easily

---

[*] Work done while doing this master thesis in Huawei.
[1]Up-to-date version at https://arxiv.org/abs/2310.13469.

identified by rule-based approaches alone.

Another highly promising approach involves leveraging large language models (LLMs) (Touvron et al., 2023a; Chen et al., 2023). Previous works have already demonstrated the effectiveness of LLMs in various tasks, including Q&A(Robinson et al., 2022), text summarization (Pilault et al., 2020), and data generation (Meng et al., 2022; Chung et al., 2023; Tang et al., 2023). However, applying an LLM to clean the target sentences poses several challenges that need to be addressed diligently:

- Comprehensive noise removal: LLMs should be capable of thoroughly cleaning the target sentence by eliminating all forms of noise, including removing semantically meaningless emojis, translating emojis with semantic content into words, correcting misspellings, etc.

- Semantic preservation: A cleaned target sentence should retain similar semantic information as the original noisy target sentence.

- Alignment with the source sentence: The cleaned target sentence should convey the same intended meaning as the noisy source sentence, ensuring accurate translation and faithful representation of the original content.

In this paper, we propose to apply GPT-3.5 (OpenAI, 2021) to clean the noisy target sentences in the MTNT dataset. Our approach addresses the challenges of noise removal, semantic preservation, and alignment with the source sentence. Inspired by the success of prompting methods (Brown et al., 2020; Wei et al., 2022; Schick et al., 2023), we design few-shot prompts to guide GPT-3.5 to clean the noisy target sentences in three ways: (1) Utilizing information from both the source and target side for cleaning, (2) Cleaning the target sentence independently, and (3) Generating a clean target sentence by translating the noisy source sentence. Through a comprehensive analysis, we demonstrate the effectiveness of our cleaning methods, particularly with methods (1) and (3). These methods successfully remove emojis, while considering their semantic meaning, and also rephrase slang, jargon, and profanities appropriately. The resulting datasets, named C-MTNT, exhibit significantly reduced levels of noise compared to the original sentences, while still preserving their semantic integrity.

| Noise Type | Example |
|---|---|
| Spelling/ typographical errors | "across" → "accross", "receive" → "recieve" |
| Word omission/ insertion/ repetition | "I never" → "I never never" |
| Grammatical errors | "a ton of" → "a tons of" |
| Spoken language | "want to" → "wanna", "going to" → "gonna" |
| Internet slang | "to be honest" → "tbh", "shaking my head" → "smh" |
| Capitalization | "Reddit"→ "reddit" |
| Dialects | African American Vernacular English, Scottish |
| Code switching | "This is so cute" → "This is so kawaii" |
| Jargon | On Reddit: "upvote", "downvote", "sub", "gild" |
| Profanities/slurs (sometimes masked) | "f*ck", "sh*" |

Table 1: Noise types and examples from MTNT.

Furthermore, our research highlights another remarkable potential of LLMs in generating high-quality parallel data with limited monolingual resources. This finding has significant implications for low-resource domains and languages, where acquiring parallel corpora is often challenging.

After the cleaning, we conduct human and GPT-4 (OpenAI, 2023) evaluations. Both evaluations draw the same conclusion that LLM has a great capability for such a cleaning task, especially with method (1).

In the end, we conduct comprehensive experiments to train NMT models on some noisy training datasets, and evaluate them on different evaluation sets, including clean benchmarks and the benchmark constructed with a rule-based noise removal method. C-MTNT consistently demonstrates better noise evaluation manner on the NMT models.

To the best of our knowledge, this is the first study to apply LLMs in the context of cleaning data, specifically addressing the challenges outlined above. Our findings highlight the potential of leveraging advanced language models for data-cleaning tasks and emphasize C-MTNT as a valuable resource for evaluating the NMT model in real-world scenarios.

## 2 Data Generation and Cleaning

The primary objective of this study is to harness the capabilities of LLMs in effectively removing noise and generating parallel language datasets to evaluate the robustness of NMT models against noisy input. In this section, we will give an overview of our data source MTNT and delineate our approach to LLM interaction.

### 2.1 MTNT Dataset

MTNT (Michel and Neubig, 2018), which incorporates noisy comments gathered from Reddit alongside professionally sourced translations, has emerged as a benchmark for assessing the performance of NMT models when exposed to noisy

input. This dataset was developed in response to the scarcity of publicly available parallel corpora containing naturally occurring noise. It encompasses three distinct languages, namely English, French, and Japanese, and offers parallel data for two language pairs: English-French and English-Japanese. As shown in Table 1, we present a comprehensive analysis of noisy types within MTNT. Notably, these noise types manifest in both source and target sentences, which is not expected since we want to evaluate the ability of an NMT model to translate a noisy source sentence into a clean target sentence. Consequently, here we devised an approach aimed at cleaning the target sentences in MTNT to enhance its assessment capability.

## 2.2 Approach

Our approach to cleaning MTNT entails meticulous consideration of various available settings and resources. It includes the assessment of its effectiveness in scenarios where bilingual resources are accessible, as well as the investigation of its feasibility in cases where only source or target data is available. To explore the capabilities of LLM, we incorporate three methods specifically tailored to different data scenarios, accounting for the varying availability of language resources.

- **Bilingual** cleaning: This approach involves providing both noisy source and target samples as input, with the focus on cleaning the target sample while preserving alignment with the source sample.

- **Monolingual** cleaning: In this approach, a noisy target sample is given as input, and a clean target sample is generated as output. It demonstrates the ability of LLM to clean sentences without relying on the original source sample that may contain excessive noise.

- **Translation**: This method generates new parallel data by taking a noisy source sample as input and producing a clean target sample as output. It showcases LLM's capability of noise ignorance.

**Chain-of-thought prompting.** Inspired by the recent achievements of prompting methods (Brown et al., 2020; Schick et al., 2023), we craft a set of few-shot examples that incorporate a coherent chain of thought (Wei et al., 2022). These examples serve to facilitate the model's comprehension

Figure 1: Prompt design for bilingual cleaning. In practice, the full prompt includes four examples. {src} and {tgt} indicate the source and target languages, respectively. Important parts are highlighted. Full examples, task descriptions and requests for all three proposed methods are in Appendix A, B and C. The other API inputs stay the same.

of diverse inputs and their corresponding handling strategies. Based on the optimal performance of four-shot examples (Brown et al., 2020), we manually curate a collection of four-shot examples for each method. The full list of examples used in our approach can be found in Appendix A.

**Prompt design.** As shown in Figure 1, each prompt consists of multiple components. The call follows a specific layout, starting with a brief description of the context, and providing essential information about the samples, domain, and language. This is followed by a concise formulation of the method: (1) Bilingual cleaning of target samples; (2) Monolingual cleaning of language samples; (3) Data generation through translation. Next,

we present a framework for the desired output, followed by the few-shot examples for each method, which include the input example, chain-of-thought reasoning for the output, and the example output itself. Finally, we insert the input sample(s) and request the desired output.

**Language model.** The prompts are utilized to interact with OpenAI's GPT through API calls. Specifically, we use the original GPT-3.5 (text-davinci-003) variant (OpenAI, 2021). In this way, we want to show that some publicly released pre-trained LLMs, like Llama 2 (Touvron et al., 2023b), might also have this ability.

**Semantic similarity.** To ensure that the text generated by our approach maintains the original meaning without unintentional hallucinations (Mündler et al., 2023), we set a threshold based on LASER (Artetxe and Schwenk, 2019). LASER is language agnostic, allowing us to compare samples not only within the same language but also across different languages. We measure the cosine similarity between the representations of original and cleaned sentences, as shown in Equation 1. (For bilingual and monolingual cleaning, the original sentence is the noisy target sentence. For translation, the original sentence is the noisy source sentence.)

$$\text{sim}(e_1, e_2) = \frac{e_1 \cdot e_2}{\|e_1\|_2 \|e_2\|_2} \quad (1)$$

where $e_1$ and $e_2$ are representations of original and newly generated sentences, respectively. Based on previous work on sentence embedding similarity (Okazaki and Tsujii, 2010; Xia et al., 2015), we set a threshold for the LASER score as 0.7. This threshold is selected to strike a balance between preserving the meaning of sentences and allowing sufficient variations. If $\text{sim}(e_1, e_2) < 0.7$, we repeat the API call but include a notice in the request: "Please ensure that your response accurately reflects the meaning of the original input sentence.", to ensure that the meaning of the new sentence aligns closely with the original one. This process continues until $\text{sim}(e_1, e_2) \geq 0.7$ or reaching the maximum number of iterations of 10.

## 3 Analysis on C-MTNT

In this section, we analyze the generated data to evaluate its quality and suitability as a noise evaluation benchmark. We compare our method to a rule-based baseline and quantitatively assess the level of noise present in the target sentences. In addition, we also compare the new target samples from C-MTNT to the original ones by evaluating their semantic similarity.

### 3.1 Baseline

In addition to our LLM approach, we utilize the language_tool_python module[2], which is an open-source grammar tool used for spelling, grammar, and overall language correction. With this rule-based baseline, we want to determine the performance gap between the rule-based method and our LLM-based approaches.

### 3.2 Quantitative Noise Measurement

We focus on several quantifiable noise types to measure the amount of noise in a set of samples and obtain an objective overview. These types are present in both the source and target sentences of MTNT, including spelling and grammatical errors, emojis, internet slang, and profanities.

We apply the language_tool_python toolkit[2] to measure the misspellings and grammatical errors. To count the occurrences of emojis in the sentences, we use the emoji library[3]. For detecting profanities within the text, we employ better_profanity[4] for English profanities, and profanity_check[5] for French and Japanese profanities. As there are no available libraries for detecting internet slang, we compile lists of popular internet slang for each language from the past ten years.

As shown in Table 2, we contend that the LLM methods possess the capability to simulate natural language as it appears in clean benchmarks such as Newstest2014[6], TED (Pryzant et al., 2018), KFTT (Neubig, 2011), and JESC (Cettolo et al., 2012), thereby generating clean target sentences. While conventional language correction tools excel in rectifying spelling and grammatical errors, they are inadequate in effectively eliminating or paraphrasing slang, profanities, or emojis. Conversely, the LLM methods demonstrate proficiency in addressing such language phenomena, as also evidenced by some samples in Appendix D. As a result, the target sentences in C-MTNT exhibit significantly less noise compared to MTNT, leveling the cleanliness of the reference benchmarks.

---

[2]https://github.com/jxmorris12/language_tool_python
[3]https://github.com/carpedm20/emoji
[4]https://github.com/snguyenthanh/better_profanity
[5]https://github.com/vzhou842/profanity-check
[6]http://www.statmt.org/wmt15/test.tgz

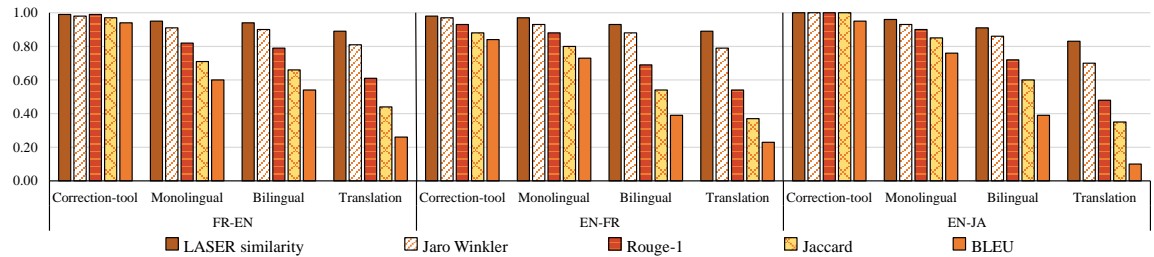

Figure 2: Semantic similarity between noisy (originally from MTNT) and cleaned English, French, and Japanese sentences. The detailed numbers for each bar plot are in Appendix E.

| Lang. | Eval. Set | Spell./Gram. | Emojis | Slang | Profanities |
|---|---|---|---|---|---|
| EN | Newstest2014 | 0.415 | 0.000 | 0.571 | 0.173 |
| | MTNT | 1.712 | 0.031 | 0.816 | 0.616 |
| | Correction-tool | **0.112** | 0.031 | 0.818 | 0.618 |
| | Bilingual | 0.687 | **0.000** | **0.584** | 0.533 |
| | Translation | 0.798 | 0.019 | 0.721 | 0.509 |
| | Monolingual | 0.748 | 0.019 | 0.716 | **0.399** |
| FR | Newstest2014 | 2.878 | 0.000 | 0.133 | 0.431 |
| | MTNT | 7.125 | 0.227 | 2.225 | 5.522 |
| | Correction-tool | **0.100** | 0.227 | 2.139 | 5.508 |
| | Bilingual | 0.552 | 0.016 | **0.691** | **0.535** |
| | Translation | 0.950 | 0.048 | 0.707 | 0.545 |
| | Monolingual | 0.455 | **0.000** | 0.715 | 0.551 |
| JA | TED | 0.049 | 0.000 | 3.493 | 3.879 |
| | KFTT | 0.011 | 0.000 | 0.486 | 4.269 |
| | JESC | 0.036 | 0.103 | 7.881 | 12.084 |
| | MTNT | 0.051 | 0.051 | **0.794** | **5.682** |
| | Correction-tool | **0.000** | 0.051 | **0.794** | 5.684 |
| | Bilingual | 0.052 | 0.012 | 1.033 | 5.783 |
| | Translation | 0.053 | 0.041 | 1.119 | 5.873 |
| | Monolingual | 0.041 | **0.006** | 0.991 | 5.874 |

Table 2: Noise frequency per 100 tokens in the target sentences of the evaluation sets. The **best scores** are highlighted. Results of Newstest2014, TED, KFTT, and JESC only serve as baselines from clean benchmarks.

Notable is the lower performance in the generated Japanese target sentences. We attribute this to two factors: insufficient capture of slang and profanities, and the known variations in performance of GPT-4 (OpenAI, 2023) across different languages (Koubaa, 2023). GPT-4 performs much worse on Japanese tasks compared to English tasks. A similar performance discrepancy is expected with GPT-3.5 (OpenAI, 2021).

### 3.3 Meaning Preservation

Our second objective is to preserve the original meaning during cleaning. We apply multiple metrics to measure the sentence similarity, including LASER (Artetxe and Schwenk, 2019), BLEU score (Papineni et al., 2002), Rouge-1 score (Lin, 2004), Jaro Winkler distance (Jaro, 1989), and Jaccard score (Hamers et al., 1989).

Figure 2 illustrates similarity scores across dif-

ferent language pairs, revealing distinct deviations among different methods. These deviations stem from different input data: bilingual, monolingual target, and monolingual source. The translation method exhibits the lowest similarity score between original and clean sentences. In contrast, the monolingual method shows the minimal deviation between original and clean sentences, while the bilingual method falls in between. We argue that the larger deviation from the bilingual and translation is mainly from rephrasing and word reordering (see Appendix D for detailed samples). Despite these variations, all methods retain a substantial portion of the original semantic structure.

Notably, similarity scores from the correction tool are the highest for all metrics among all methods, since this rule-based method can only clean or remove noise but lacks the ability to rephrase challenging noise types like slang, emojis, and profanities (see Table 2 for its results on slang, profanities, and emojis). Most cleaned sentences stay very similar to the original noisy ones. Complemented by the findings in Section 3.2, our cleaning methods show their impressive ability in reducing noise while preserving semantic similarity.

### 3.4 Human and GPT-4 Evaluations

Apart from evaluating C-MTNT by measuring its noise amount and its semantic preservation, here we conduct human and GPT-4 (OpenAI, 2023) evaluations.

Due to the limited research budget, we only conducted the human evaluation with the help of the first four authors of this paper on some sampled sentences instead of all sentences. 100 sentences from C-MTNT Fr→En are sampled to generate three files. These three files are about binary comparisons of bilingual vs. monolingual, bilingual vs. translation, and monolingual vs. translation. The order of sentences, including their indexes

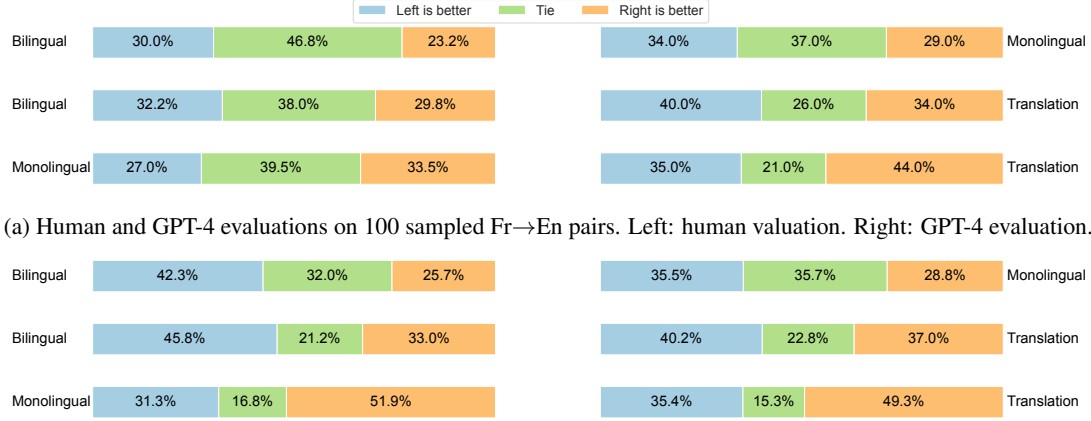

(a) Human and GPT-4 evaluations on 100 sampled Fr→En pairs. Left: human valuation. Right: GPT-4 evaluation.

(b) GPT-4 evaluation on all samples. Left: En→Fr. Right: Fr→En.

Figure 3: Human and GPT-4 evaluations on C-MTNT. Overall, the Bilingual cleaning method results in the most cleaned target sentences.

and which cleaning method comes first, is randomly shuffled. There is no chance for the annotator to guess which sentence corresponds to which method, and which file corresponds to the comparison between which two methods. Notably, we prefer binary comparison to ranking over three methods, since it's easier for human annotators. In addition, the cleaned sentences from the correction tool are excluded, since they are too easy to be beaten. Four annotators are asked to give their preferences for each comparison, based on our three criteria of comprehensive noise removal, semantic preservation, and alignment with the source sentence. If both sentences show a similar level of noise, they are asked to give a "Tie".

We also prompt GPT-4 (See Appendix G for the prompt) on the same sampled sentences to check whether GPT-4 draws a similar conclusion. As shown in Figure 3a, human and GPT-4 evaluations share a similar preference: bilingual > translation > monolingual. Compared to GPT-4, human annotators prefer to vote for "Tie". We argue the main reason is that most cleaned sentences are very similar with a low level of noise, which further justifies the effectiveness of our proposed methods. Since human annotators and GPT-4 share a similar preference, we further evaluate all C-MTNT sentences with GPT-4 in Figure 3b. The conclusion is similar to the above discussion, i.e. bilingual > translation > monolingual.

## 4 Machine Translation Experiments

In this section, we further investigate the suitability of C-MTNT as a benchmark to evaluate NMT model's robustness against noisy input. Let's re-emphasize our expected NMT model: Irrespective of whether the source sentence is clean or noisy, the model has the ability to generate a coherent target sentence that is free of errors or distortions.

We first mimic the real-world scenario to train a set of NMT models on datasets that contain both noisy and clean source sentences but with only clean target sentences, then evaluate these models on C-MTNT and other benchmarks.

### 4.1 Model and Training Details

All models are trained with the fairseq toolkit (Ott et al., 2019). The architecture is based on the vanilla transformer, with 6 encoder and 6 decoder layers. The hidden dimension is set to 512, with 1024 for the feed-froward intermediate output. We use Adam optimizer (Kingma and Ba, 2015) with its default hyperparameters. Dropout with a probability of 0.3 and a label smoothing factor of 0.1 (Gao et al., 2020) is applied. We train all models for 20 epochs with a learning rate of $5e-4$ that is scheduled by inverse square root with 4K warmup steps, and set a batch size as 128. Subword tokenization is performed using SentencePiece (Kudo and Richardson, 2018) with BPE subwords. For all languages, we use a vocabulary size of 16K without sharing embeddings.

### 4.2 Training Data

For the English ↔ French translation directions, we utilize the same training data as Michel and Neubig (2018), which comprises the europarl-v7[7]

---

[7]http://www.statmt.org/europarl/

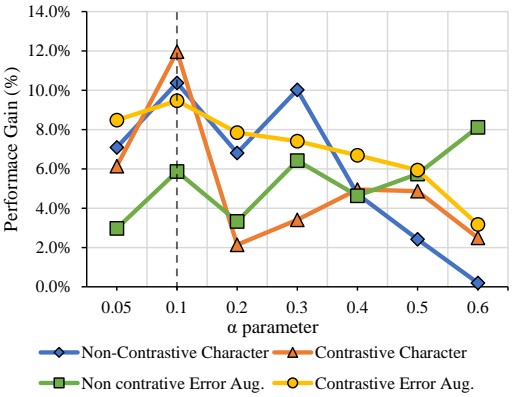

Figure 4: Performance vs. augmentation strength, $\alpha$, for models trained with different augmentation strategies on the French-to-English translation direction. Scores are computed on the bilingual C-MTNT evaluation set.

and news-commentaryv10[8] corpora. The training set consisted of 2.2M samples, with 55M French tokens and 52M English tokens (non-tokenized). The WMT15 newsdiscussdev2015[9] serves as the validation set, used to select the best checkpoint. The trained models are evaluated on the C-MTNT, MTNT, and newstest2014[6] test (eval.) sets.

Regarding the English-to-Japanese translation direction, we follow the data construction approach of Michel and Neubig (2018), combining the training and validation data from the KFTT (Neubig, 2011), JESC (Pryzant et al., 2018), and TED talks (Cettolo et al., 2012) corpora. The Japanese segments in each dataset are tokenized using the MeCab library (Kudo, 2005). This results in a training set of 3.9M samples, consisting of 35M English tokens without tokenization. We use the training and dev sets associated with each corpus as the training and validation sets, respectively, and evaluate the models on MTNT, C-MTNT, and the respective test set from each corpus.

### 4.3 Data Augmentation

The above-mentioned training data are clean, contain negligible noise, and can't resemble the real-world use case. Therefore, we introduce noise with different augmentation methods to the source sentences, including character augmentation (spelling/-typographical errors, capitalization), contextual word embedding augmentation (word omission/insertion/repetition), MTNT-based error replacement (spoken language, jargon, internet slang, grammati-

cal errors), and synonym substitution (grammatical errors).

**Character augmentation (Char.)** involves character-level augmentation methods, including random or controlled techniques with a probability of 0.5 for each choice (Karpukhin et al., 2019).

**Contextual word embedding augmentation (Con.)** utilizes language models, specifically BERT (bert-base-uncased) (Devlin et al., 2019), to substitute some word embeddings with their contextual word embeddings (Kobayashi, 2018). We employ the French BERT [10] for French source sentences.

**MTNT-based error replacement (Err.)** is inspired by the symbolic strategies (Shorten et al., 2021). Errors are identified with language_tool_python, and only the most valuable ones occurring more than once are retained in a dictionary for augmentation. By replacing the correct forms with the mapped common errors, we intentionally introduce these errors into the sentence.

For **synonym substitution (Syn.)**, we employ WordNet (Miller, 1995) and NLTK (Bird et al., 2009), to randomly select and replace words with their synonyms.

These techniques are only tailored for English and French, which is also the main reason for our exclusion of Japanese-to-English direction. The source sentences $x$ are augmented with a probability of $\alpha = 0.1$, augmenting approximately 10% of the tokens in each sample. This process generates four augmented versions: $z_{ch}$, $z_c$, $z_e$, and $z_s$, representing sentence augmentation with character, contextual word embedding, error, and synonym augmentation, respectively. The selection of $\alpha = 0.1$ is based on previous works (Karpukhin et al., 2019; Wei and Zou, 2019) and our similar finding (see Figure 4). These augmented sentences are combined with the original clean sentences, resulting in four new training sets, $\{x, z_{ch}\}$, $\{x, z_c\}$, $\{x, z_e\}$, and $\{x, z_s\}$. Each is used to train a model, capable of handling some specific types of noise.

Notably, the distribution of introduced noise is not possible to totally resemble the noise distribution in the MTNT (or C-MTNT) source sentences, since we only introduce some types of noise to each new set with a pre-defined rule, i.e. the augmentation method. This setting is desired and makes the evaluation more general and practical.

[8]http://www.statmt.org/wmt15/training-parallel-nc-v10.tgz

[9]http://www.statmt.org/wmt15/dev-v2.tgz

[10]https://github.com/stefan-it/europeana-bert

| | Eval. Set | Base. | Non-Contrastive | | | | Contrastive | | | |
|---|---|---|---|---|---|---|---|---|---|---|
| | | | Char. | Syn. | Con. | Err. | Char. | Syn. | Con. | Err. |
| FR-EN | Newstest2014 | 29.2 | $29.9_{2.4}$ | $29.1_{-0.2}$ | $28.9_{-1.1}$ | $29.2_{0.0}$ | $29.4_{0.7}$ | $29.2_{0.0}$ | $28.5_{-2.5}$ | $29.6_{1.2}$ |
| | MTNT | 23.1 | $25.0_{8.5}$ | $23.6_{2.2}$ | $23.2_{0.7}$ | $23.6_{2.4}$ | $24.8_{7.5}$ | $22.9_{-0.7}$ | $22.7_{-1.4}$ | $23.6_{2.3}$ |
| | Correction tool | 23.2 | $25.2_{8.4}$ | $23.7_{2.1}$ | $23.4_{0.7}$ | $24.3_{4.8}$ | $24.9_{7.4}$ | $23.0_{-0.7}$ | $23.0_{-0.9}$ | $23.8_{2.6}$ |
| | Bilingual | 25.3 | $\mathbf{28.3_{12.0}}$ | $26.7_{5.6}$ | $27.0_{6.9}$ | $\mathbf{26.7_{5.9}}$ | $27.9_{10.4}$ | $27.0_{6.9}$ | $\mathbf{26.1_{3.5}}$ | $27.6_{9.5}$ |
| | Translation | 26.2 | $29.0_{10.7}$ | $27.6_{5.5}$ | $27.8_{6.1}$ | $27.4_{4.6}$ | $28.6_{9.0}$ | $27.4_{4.7}$ | $26.7_{1.8}$ | $28.1_{7.4}$ |
| | Monolingual | 21.4 | $\underline{23.7_{10.8}}$ | $21.7_{1.8}$ | $21.5_{0.8}$ | $21.8_{2.2}$ | $23.0_{7.6}$ | $20.9_{-1.9}$ | $20.8_{-2.7}$ | $21.7_{1.5}$ |
| EN-FR | Newstest2014 | 30.3 | $33.1_{9.0}$ | $31.7_{4.5}$ | $31.5_{3.9}$ | $31.9_{5.3}$ | $32.6_{7.6}$ | $32.4_{6.8}$ | $32.3_{6.7}$ | $32.6_{7.4}$ |
| | MTNT | 20.1 | $22.4_{11.5}$ | $20.7_{3.1}$ | $21.2_{5.7}$ | $21.8_{8.5}$ | $22.2_{10.7}$ | $21.4_{6.5}$ | $21.1_{5.3}$ | $22.3_{11.4}$ |
| | Correction tool | 19.7 | $22.0_{11.6}$ | $20.5_{3.9}$ | $20.9_{6.0}$ | $21.4_{8.5}$ | $21.9_{11.1}$ | $21.0_{6.3}$ | $20.9_{5.7}$ | $22.0_{11.6}$ |
| | Bilingual | 19.7 | $\underline{22.1_{12.2}}$ | $20.7_{5.1}$ | $21.0_{6.9}$ | $\mathbf{21.6_{10.1}}$ | $22.0_{12.2}$ | $\mathbf{21.3_{8.2}}$ | $\mathbf{21.1_{7.4}}$ | $22.6_{15.0}$ |
| | Translation | 19.3 | $\mathbf{21.9_{13.5}}$ | $\mathbf{20.5_{6.0}}$ | $\mathbf{20.8_{7.7}}$ | $21.1_{9.2}$ | $\mathbf{21.7_{12.6}}$ | $20.9_{8.0}$ | $20.7_{7.0}$ | $\mathbf{22.5_{16.4}}$ |
| | Monolingual | 16.6 | $18.5_{11.3}$ | $17.2_{3.9}$ | $17.6_{6.3}$ | $17.9_{8.0}$ | $18.5_{11.5}$ | $17.5_{5.8}$ | $17.5_{5.6}$ | $18.8_{13.4}$ |
| EN-JA | TED | 14.4 | $14.9_{3.3}$ | $15.0_{4.1}$ | $14.2_{-1.6}$ | $14.4_{0.0}$ | $14.9_{3.8}$ | $14.4_{0.0}$ | $14.2_{-1.6}$ | $14.6_{1.5}$ |
| | KFTT | 24.9 | $25.7_{3.2}$ | $24.5_{-1.7}$ | $25.0_{0.2}$ | $24.2_{-2.8}$ | $25.5_{2.3}$ | $24.7_{-0.8}$ | $25.0_{0.2}$ | $24.6_{-1.3}$ |
| | JESC | 15.2 | $15.0_{-1.4}$ | $14.8_{-2.4}$ | $14.9_{-1.9}$ | $15.0_{1.1}$ | $15.1_{-0.6}$ | $14.8_{-2.9}$ | $14.9_{-1.9}$ | $14.9_{-1.7}$ |
| | MTNT | 8.8 | $9.0_{2.4}$ | $9.1_{3.5}$ | $9.0_{2.5}$ | $9.0_{1.7}$ | $9.0_{2.6}$ | $9.1_{2.8}$ | $9.0_{2.6}$ | $8.9_{0.8}$ |
| | Correction tool | 8.8 | $9.0_{2.5}$ | $9.1_{3.6}$ | $9.0_{2.6}$ | $8.9_{0.6}$ | $9.1_{2.7}$ | $9.1_{2.8}$ | $9.0_{2.6}$ | $8.9_{0.9}$ |
| | Bilingual | 12.6 | $\mathbf{14.0_{11.3}}$ | $13.9_{10.1}$ | $13.3_{5.5}$ | $13.1_{3.7}$ | $14.1_{11.7}$ | $13.7_{8.3}$ | $\underline{13.3_{5.5}}$ | $13.4_{6.7}$ |
| | Translation | 14.6 | $16.1_{10.2}$ | $15.9_{8.8}$ | $\mathbf{15.5_{6.1}}$ | $\mathbf{15.7_{7.3}}$ | $16.4_{12.1}$ | $16.1_{10.5}$ | $\mathbf{15.5_{6.1}}$ | $\mathbf{15.7_{7.8}}$ |
| | Monolingual | 9.3 | $10.1_{8.7}$ | $10.0_{7.7}$ | $9.7_{4.2}$ | $9.2_{-1.4}$ | $10.1_{8.2}$ | $\underline{10.1_{8.4}}$ | $9.7_{4.2}$ | $9.2_{-1.1}$ |

Table 3: BLEU scores on various evaluation sets. The subscript number is the relative performance gain $G$ (%), calculated as Equation 2. The **best** and second-best $G$ for each augmented dataset are highlighted and underlined, respectively. Results of Newstest2014, TED, KFTT, and JESC only serve as baselines from clean benchmarks.

## 4.4 Contrastive Learning

In addition to the straightforward training on newly constructed sets, we also train models with contrastive learning, which is inspired by previous works (Chen et al., 2020; Hwang et al., 2021; Hu and Li, 2022) that recognize the effectiveness of contrastive learning in improving the robustness of NLP and NMT models. By employing this method, we can analyze the performance of C-MTNT on a wider range of models trained with different approaches and settings.

For contrastive learning, the Transformer encoder takes both original $x$ and augmented $z$ source sentences as inputs and calculates the contrastive loss based on their output representations. Similar to straightforward training, we train a separate model on each set with contrastive learning. More details and experiments on contrastive training are in Appendix F.

## 4.5 Results and Analysis

Before introducing our results in Table 3 for clean benchmarks, MTNT and C-MTNT, we first want to emphasize that the BLEU scores across different benchmarks are incomparable because these benchmarks contain different evaluation sentences. Though MTNT and C-MTNT contain the same source sentences, their target sentences are different since we apply LLM to clean the target side of MTNT. Therefore, we focus on the relative performance gain:

$$G = (s_r - s_b)/s_b \qquad (2)$$

where $s_b$ is the BLEU score from the model trained only on the data without any augmentation, i.e. the training dataset in Section 4.2, and $s_r$ is the BLEU score from the model trained on the augmented dataset. If a model trained with the augmented dataset obtains higher $G$ on an evaluation set, we can say the evaluation set is an ideal noise evaluation set. The reason is: The augmented dataset mimics our expected data distribution. I.e. noise only exists in the source side. A model trained on this dataset is supposed to have the ability to translate noisy source sentences into clean target sentences. If this model obtains a high $G$ on an evaluation set, it shows that this evaluation set also fulfills the expected data distribution.

As shown in Table 3, all trained models tend to have significantly higher $G$ for bilingual and translation C-MTNT. We also show the average $G$ over four models trained with different augmented datasets in Figure 5. It is even more evident that bilingual and translation C-MTNT offers higher $G$ across all cleaning methods.

Some may argue that C-MTNT achieves a higher $G$ score because the augmented training dataset has a similar distribution to C-MTNT, making it

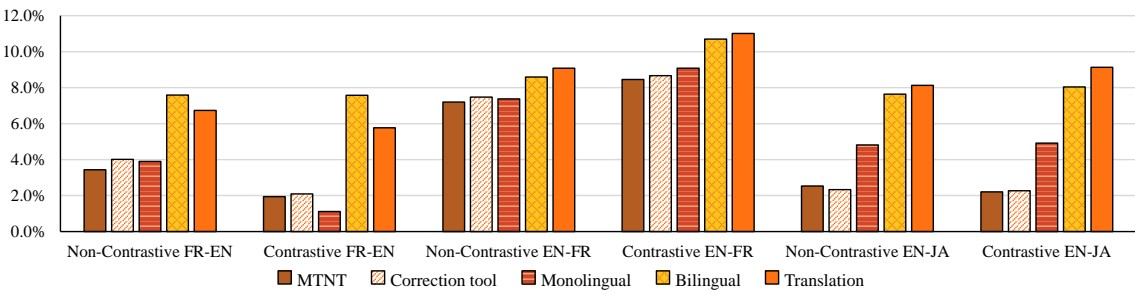

Figure 5: Average relative performance gain $G$ over four models that are trained on different augmented datasets.

easier to evaluate the trained models on C-MTNT. However, this argument is not valid for two reasons: (1) Bilingual and translation C-MTNT consistently offer higher $G$ across all models trained with different augmented datasets (see Table 3). It's almost impossible to intentionally make every augmented dataset has a similar distribution as C-MTNT where the source sentence contains natural noise; (2) Monolingual C-MTNT offers lower $G$, sometimes even lower than MTNT and the benchmark constructed from the correction tool. This shows that cleaning with a LLM doesn't always work. It's better to have guidance, like guidance from a source sentence for the bilingual and translation methods. According to our observation, if we clean the noisy target sentence in a monolingual way without any guidance, LLM tends to introduce extra information or delete important information, which hurts translation because the cleaned target sentence doesn't align well with the source sentence. In sum, C-MTNT generated by bilingual and translation methods shows its superiority as a noise evaluation benchmark, encouraging a NMT model to translate a noisy source sentence to a clean target sentence.

## 5   Related Work

The application of LLMs, such as GPT-3.5 (text-davinci-003) (OpenAI, 2021), in downstream tasks has garnered significant attention and led to extensive research in this field (Wang et al., 2022; Schick et al., 2023). Researchers have conducted comprehensive investigations to explore the capabilities of LLMs, and built upon their various applications (Coyne and Sakaguchi, 2023; Mündler et al., 2023; Jiao et al., 2023).

Researchers have also observed the potential of LLMs to generate high-quality data, leading to a focus on expanding existing datasets, augmenting data, or generating entirely new data (Meng

et al., 2022; Yoo et al., 2021; Chung et al., 2023). These efforts have helped address the issue of data scarcity in various domains.

However, it is crucial to note that the aforementioned research works lack at least one of the two novel aspects addressed in this paper. Firstly, our research focuses on evaluating the robustness of NMT models to real-world noise introduced by human users. Secondly, we explore the generation or cleaning of parallel data specifically for NMT purposes. These unique aspects of robustness evaluation and parallel data generation/cleaning contribute to the existing literature in a novel way.

## 6   Conclusion

In this work, we propose three methods to apply LLM to clean a noisy MT benchmark, MTNT, where natural noise exists in both source and target sentences. We aim to clean the target side of MTNT to make it more suitable as a benchmark in evaluating NMT models' robustness against noisy input. With a meticulous design of some few-shot prompts, we guide GPT to clean the noisy target sentence with only the noisy source sentence, only the noisy target sentence, or both noisy source and target sentences. By measuring the noise frequency in the cleaned target sentences, measuring the semantic similarity between noisy and cleaned target sentences, and evaluating with human annotators and GPT-4, we show that our proposed methods can effectively remove natural noise with LLM, while preserving the semantic structure. Our further investigation of the newly created benchmark, C-MTNT, on some trained models also shows its effectiveness as a noise evaluation benchmark for NMT models.

## 7   Limitations

Despite the contributions and potential benefits of our research, there are several limitations that

should be acknowledged. Firstly, our approach relies on the use of pre-trained LLMs for data cleaning and generation. While LLMs have shown promising capabilities, they are not immune to biases and limitations present in the training data. As a result, our proposed dataset may still contain biases similar to those found in the original MTNT dataset, even after our efforts to mitigate them.

Furthermore, our assessment of the robustness of NMT models against noisy input relies on the utilization of C-MTNT, which is created using our proposed methodology, and MTNT. While C-MTNT offers valuable insights into the performance of NMT models, it is crucial to acknowledge that it may not comprehensively represent all potential sources of noise encountered in real-world settings. Human-generated noise exhibits variability and contextual dependencies, and our dataset may not encompass the entire spectrum of noise that NMT models may face during actual deployment. The same can be said for MTNT.

Additionally, our research focuses on evaluating the robustness of NMT models in specific language directions, namely English ↔ French and English → Japanese. While these directions provide valuable insights, generalizing the findings to other language pairs should be done with caution. Different languages may exhibit unique linguistic characteristics, which can influence the performance and robustness of NMT models. Therefore, further research is needed to investigate the generalizability of our findings across a broader range of languages and translation directions.

In summary, while our research contributes to the assessment of NMT model robustness and the generation of high-quality evaluation datasets, it is important to recognize the limitations associated with biases in LLMs, the potential incompleteness of our dataset, and the need for further investigation into different language pairs.

## 8 Ethical Considerations

The utilization of pre-trained LLMs in natural language processing tasks, including data generation and machine translation, presents several ethical considerations that must be carefully examined. In this section, we discuss the ethical implications associated with the use of LLMs and highlight the potential biases that may arise in C-MTNT.

### 8.1 Biases in Pre-trained Large Language Models

Pre-trained LLMs, such as GPT-3.5, are trained on vast amounts of internet text, which inevitably introduces biases present in the training data. These biases can manifest in different forms, including but not limited to cultural, gender, racial, and political biases. The models can inadvertently reproduce and amplify these biases when generating new content or translating text.

It is crucial to acknowledge that biases present in LLMs can influence the quality and fairness of the generated data, potentially perpetuating societal inequalities and reinforcing existing stereotypes. The responsible use of LLMs requires diligent examination and mitigation of these biases to ensure equitable outcomes and avoid further marginalization or harm to underrepresented groups.

### 8.2 Mitigating Biases in Data Generation

While we employ LLMs for data cleaning and generation in our proposed dataset, it is essential to note that biases similar to those in MTNT may be present in the generated data. Despite efforts to mitigate biases, the LLMs may not fully capture the complexities and nuances of language, leading to potential biases in the generated sentences.

We carefully evaluated the generated data for any biased content and took steps to minimize biased outputs. Additionally, we encourage the involvement of diverse annotators and domain experts during the evaluation and curation of the dataset to ensure a broader perspective and mitigate the influence of individual biases. We also encourage translators and reviewers who are well-versed in the target languages and cultural nuances to ensure the translations accurately reflect the intended meaning while avoiding biased or offensive content. Moreover, we actively seek feedback from the affected communities and stakeholders to address any concerns and rectify biases that might arise.

## Acknowledgements

We thank all EMNLP reviewers for their great feedback. The first author, Quinten Bolding, finished this work while doing his thesis at Huwai Amsterdam Research Center. This work was supported by Huawei's infrastructure. The thesis supervisor, Baohao Liao, is funded in part by the Netherlands Organization for Scientific Research (NWO) under project number VI.C.192.080.

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

## A Few-Shot Examples

**Bilingual cleaning for FR-EN.** For the bilingual cleaning in the French-to-English translation direction, we use the following examples:
```
Input:  "Jss tro contenteee!",
"Im soooo happyyyy!  ☺ "
Desired output:  "I'm so happy!
☺ "
Reasoning:  The noisy French
sentence uses abbreviations
("Jss" for "Je suis") and
excessive letters in the
word "contenteee." The clean
English sentence replaces the
abbreviations and removes the
extra letters to convey the same
meaning clearly.

Input:  "Tkt, c trooo bi1!  😎
", "Don't worry, it's awesooome!
😎 "
Desired output:  "Don't worry,
it's great!"
Reasoning:  The noisy French
sentence uses internet slang
("Tkt" for "T'inquiète") and
excessive letters in the word
"trooo." The clean English
sentence replaces the slang with
a more standard expression and
removes the extra letters for
better comprehension.

Input:  "Jvoudré vnir avc vs!
☺ ", "I wud luv 2 cum wth u guys!
☺ "    Desired output:  "I would
love to come with you!"
Reasoning:  The noisy French
sentence uses abbreviations
("Jvoudré" for "Je voudrais")
and non-standard spelling
("vnir" for "venir").  The clean
English sentence replaces the
abbreviations and corrects
the spelling for clearer
communication.

Input:  "Ct troo bi1, jkiff
trp!♡ ", "It's sooo gre8, I luv
it!♡ "
Desired output:  "It's amazing, I
```
```
love it!"
Reasoning:  The noisy French
sentence uses non-standard
spelling ("troo" for "très") and
excessive letters in the words
"bi1" and "jkiff." The clean
English sentence corrects the
spelling and removes the extra
letters for a more natural and
concise expression.
```

**Bilingual cleaning for EN-FR.** For the bilingual cleaning in the English-to-French translation direction, we use the following examples:
```
Input:  "I'm sooooo happyyyy!☺ ",
"Je suis tellement heureuxxxx !☺
"
Desired output:  "Je suis
tellement heureux!"
Reasoning:  The noisy French
sentence contains excessive
letters in the word "heureux"
and an unnecessary exclamation
mark.  The clean French sentence
removes the extra letters and the
exclamation mark to convey the
same message accurately.

Input:  "Can't wait to see
youuuu!  ☺ ", "J'ai trooooop hâte
de te voiiiiir !  ☺ "
Desired output:  "J'ai tellement
hâte de te voir!"
Reasoning:  The noisy French
sentence includes excessive
letters in the words "trooop"
and "voiiiiir." The clean French
sentence removes the extra
letters to maintain the same
meaning more concisely.

Input:  "Let's grab a bite
laterrrr!  🍜 🍔 ", "Allons
manger un morceau plu tarrrrd!
🍜 🍔 "
Desired output:  "Allons manger
un morceau plus tard!"
Reasoning:  The noisy French
sentence has excessive letters
in the words "plu" and "tarrrrd"
and unnecessary fast food emojis.
```

The clean French sentence removes the extra letters and the emojis while maintaining the same meaning.

Input: "This movie is amaziiing! ⛺ ♡ ", "Ce film est troooop géniaaaaal! ⛺ ♡ "
Desired output: "Ce film est tellement génial!"
Reasoning: The noisy French sentence contains excessive letters in the words "troooop" and "géniaaaaal" and unnecessary fire and heart emojis. The clean French sentence removes the extra letters and the emojis to convey the same message accurately.

**Bilingual cleaning for EN-JA.** For the bilingual cleaning in the English-to-Japanese translation direction, we use the following examples:
Input: "I'm soooo happyyyy! ☺ ", "すっごーく嬉しい！☺ "
Desired output: "すごく嬉しいです！"
Reasoning: The noisy Japanese sentence uses excessive elongation in the word "すっごー く" and includes an unnecessary exclamation mark. The clean Japanese sentence removes the excessive elongation and uses a more polite form to convey the same meaning accurately.

Input: "Can't wait to see youuuu! ☺ " Desired output: "会えるのが楽しみです！☺ "
Reasoning: The noisy Japanese sentence includes excessive elongation in the word "よぉぉぉ" and an unnecessary exclamation mark. The clean Japanese sentence removes the excessive elongation and uses a more polite form for a clearer and more appropriate expression.

Input: "Let's grab a bite laterrrr! ⛲ 🍱 ", "後 で 軽 く 食 べよっかあああ！⛲ 🍱 "
Desired output: "後でちょっと食べましょう！"
Reasoning: The noisy Japanese sentence includes excessive elongation in the word "よ っ か ぁ ぁ ぁ" and unnecessary fast food emojis. The clean Japanese sentence removes the excessive elongation and provides a more polite and appropriate phrase to convey the same meaning.

Input: "This movie is amaziiing! ⛺ ♡ ", "この映画はすっごいいぃぃ ⛺ ♡ "
Desired output: "この映画は素晴らしいです！"
Reasoning: The noisy Japanese sentence uses excessive elongation in the word "す っ ご いぃぃぃ" and includes unnecessary fire and heart emojis. The clean Japanese sentence removes the excessive elongation and provides a more appropriate and accurate expression for the same meaning.

**Translation for FR-EN.** For the generative translation method in the French-to-English translation direction, we use the following examples:
Input: "Heyyy, ça va trop biennn! ☺ Jsuis trop hypeééé pour ce soir! ♡ "
Desired output: "Hey, I'm doing great! I'm so excited for tonight!"
Reasoning: The noisy sentence contains excessive letters in words and emojis. The clean sentence removes the extra letters and emojis to convey the same message more clearly.

Input: "OMG jpeux pas croire, c'est trooop ouf! ⛺ ⛺ "
Desired output: "Oh my God, I can't believe it, it's so amazing!"

Reasoning: The noisy sentence uses internet slang ("OMG," "trooop," "ouf") and excessive punctuation ("!!"). The clean sentence replaces the slang with more standard expressions and removes the excessive punctuation for better comprehension.

Input: "Mdr t'es trop marrant, tu me fais tp rire 😊 😊 "
Desired output: "Haha, you're so funny, you make me laugh a lot."
Reasoning: The noisy sentence contains internet slang ("Mdr," "trop," "tp") and a laughing emoji. The clean sentence replaces the slang with more common expressions and removes the emoji for a more formal and clear communication.

 Input: "Héé, on se voit au restau tout de suite?"
Desired output: "Hey, can we meet at the restaurant right away?🍜 📷 "      Reasoning: The noisy sentence has intentional misspellings ("Héé," "restau") and fast food emojis. The clean sentence corrects the spellings and removes the emojis to convey the same message accurately.

**Translation for EN-FR.** For the generative translation method in the English-to-French translation direction, we use the following examples:
Input: "Heyy, what's up? 😊 I'm sooo exicteddd to go out tonight! ♡ "
Desired output: "Salut, quoi de neuf ? Je suis tellement excité(e) de sortir ce soir !"
Reasoning: The noisy sentence contains excessive letters in words and emojis. The clean sentence removes the extra letters and emojis to convey the same message more clearly in French.

Input: "OMG I can't even rn, this party is gonna be LIT AF! 🔥 🔥 "
Desired output: "Mon Dieu, je ne peux pas le croire, cette fête va être incroyable !"
Reasoning: The noisy sentence uses internet slang ("OMG," "rn," "LIT AF") and abbreviations. The clean sentence replaces the slang with more standard expressions and removes the abbreviations for better comprehension in French.

Input: "lol ur so funny, u always make me laugh 😊 😊 "
Desired output: "lol tu es tellement drôle, tu me fais toujours rire."
Reasoning: The noisy sentence contains internet slang ("lol," "ur") and a laughing emoji. The clean sentence replaces the slang with more common expressions and removes the emoji for a more formal and clear communication in French.

Input: "Heyy, wanna grab some fud later? 🍜 📷 "
Desired output: "Hey, tu veux qu'on aille manger plus tard ?"
Reasoning: The noisy sentence has intentional misspellings ("Heyy," "fud") and fast food emojis. The clean sentence corrects the spellings and removes the emojis to convey the same message accurately in French.

**Translation for EN-JA.** For the generative translation method in the English-to-Japanese translation direction, we use the following examples:
Input: "Heyyy, what's up? 😊 I'm sooo excited to go out tonight! ♡ "
Desired output: "やぁやぁ、元気？今夜出かけるのがすっごく楽しみ！"
Reasoning: The noisy sentence

contains excessive letters in words and emojis. The clean sentence removes the extra letters and emojis to convey the same message more clearly in Japanese.

Input: "OMG I can't even rn, this party is gonna be LIT AF! ⚱ ⚱ " Desired output: "ええっ、マジで？！このパーティは最高に盛り上がるぞ"
Reasoning: The noisy sentence uses internet slang ("OMG," "rn," "LIT AF") and abbreviations. The clean sentence replaces the slang with more standard expressions and removes the abbreviations and emojis for better comprehension in Japanese.

Input: "lol you're so funny, you always make me laugh 😊 😊 "
Desired output: "笑 お前、ホント面白いな。いつも笑わせてくれるんだよ"
Reasoning: The noisy sentence contains internet slang ("lol") and a laughing emoji. The clean sentence replaces the slang with more common expressions and removes the emoji for a more natural and clear communication in Japanese.

Input: "Heyyy, wanna grab some food later? 🍜 🍱 "
Desired output: "やぁやぁ、後で食べ物でも買っていかない？"
Reasoning: The noisy sentence has intentional misspellings ("Heyyy") and fast food emojis. The clean sentence corrects the spellings and removes the emojis for a more natural and clear communication in Japanese.

**Monolingual cleaning for FR-EN.** For the monolingual cleaning in the French-to-English translation direction, we use the following examples:
Input: "Heyy, what's up? 😊 I'm sooo exicteddd to go out tonight! ♡ "
Desired output: "Hey, what's up? I'm so excited to go out tonight!"
Reasoning: "The noisy sentence contains excessive letters in words and emojis. The clean sentence removes the extra letters and emojis to convey the same message more clearly."

Input: "OMG I can't even rn, this party is gonna be LIT AF! ⚱ ⚱ "
Desired output: "Oh my God, I can't even right now, this party is going to be awesome!"
Reasoning: "The noisy sentence uses internet slang ('OMG', 'rn', 'LIT AF') and abbreviations. The clean sentence expands the slang and abbreviations for better comprehension."

Input: "lol ur so funny, u always make me laugh 😊 😊 "
Desired output: "Laugh out loud, you're so funny. You always make me laugh."
Reasoning "The noisy sentence contains internet slang ('lol', 'ur') and a laughing emoji. The clean sentence replaces the slang with its full form and removes the emoji for a more formal expression."

Input: "Heyy, wanna grab some fud later? 🍜 🍱 "
Desired Output "Hey, do you want to grab some food later?"
Reasoning: "The noisy sentence has intentional misspellings ('Heyy', 'fud') and food emojis. The clean sentence corrects the spellings and removes the emojis to convey the same message accurately."

**Monolingual cleaning for EN-FR.** For the monolingual cleaning in the English-to-French

translation direction, we use the following examples:

Input: "Saluttt, ça va? ☺ Je suiis trp excitéééé pr sortir ce soiiiir! ♡ "
Desired output: "Salut, ça va? Je suis trop excité pour sortir ce soir!"
Reasoning: The noisy sentence contains excessive letters in words and emojis. The clean sentence removes the extra letters and emojis to convey the same message more clearly.

Input: "Tkt, j'te dm dans 2min, ok? ☺ "
Desired output: "T'inquiète, je te donne des nouvelles dans 2 minutes, d'accord?"
Reasoning: The noisy sentence uses internet slang ('Tkt', 'j'te', 'dm') and abbreviations. The clean sentence expands the slang and abbreviations for better comprehension.

Input: "Mdrr t'es tro drol, tu m'fais tp rire ☺ ☺ "
Desired output: "Mort de rire, tu es vraiment drôle, tu me fais trop rire."
Reasoning: The noisy sentence contains internet slang ('Mdrr', 'tro', 'tp') and a laughing emoji. The clean sentence replaces the slang with its full form and removes the emoji for a more formal expression.

Input: "Héé, on se retrouve au mcdo plutar? 🍽 ☕ "
Desired output: "Hé, est-ce qu'on peut se retrouver au McDonald's plus tard?"
Reasoning: The noisy sentence has intentional misspellings ('Héé', 'plutar') and fast food emojis. The clean sentence corrects the spellings and removes the emojis to convey the

same message accurately.

**Monolingual cleaning for EN-JA.** For the monolingual cleaning in the English-to-Japanese translation direction, we use the following examples: Input: "元気っすか？☺ めっちゃ楽しみだぜ~♡ "
Desired output: "元気ですか？とっても楽しみですね！"
Reasoning: The noisy sentence contains informal language ("っすか" instead of "ですか") and excessive use of " " at the end. The clean sentence removes the informal elements and expresses the same meaning more formally.

Input: "めっちゃおいしーい！LOL☺ "
Desired output: "とってもおいしい！笑"
Reasoning: The noisy sentence includes the use of "めっちゃ" (a casual intensifier) and the English acronym "LOL." The clean sentence removes the casual intensifier and replaces "LOL" with the Japanese equivalent "笑" (meaning "laugh").

Input: "アハハ、超おもろい！☺ "
Desired output: "笑、とても面白い！"
Reasoning: The noisy sentence uses "アハハ" (a casual laughter expression) and an emoji. The clean sentence replaces "アハハ" with the more standard "笑" and removes the emoji.

Input: "よっしゃ、待ち合わせまつか？🍽 ☕ "
Desired output: "よし、待ち合わせしましょうか？"
Reasoning: The noisy sentence contains a misspelling ("まつか" instead of "ましょうか") and fast food emojis. The clean sentence corrects the spelling and removes the emojis while maintaining the same meaning.

## B Task Descriptions

The task description for the bilingual cleaning method is as follows:

```
Your task is to clean the given
{tgt} sentence.  You will receive
two sentences as input:  the
{src} sentence containing noise,
and the translated tgt version
of that sentence, also containing
noise.  Your task is to clean
only the {tgt} sentence and
return it as output.
```

The task description for the generative translation method is as follows:

```
Your task is to translate the
given noisy {src} sentence to the
correct {tgt} version, thereby
removing all noise.  You will
only return the clean {tgt}
sentence as output.
```

The task description for the monolingual cleaning method is as follows:

```
Your task is to clean the {tgt}
sentence that you will receive
as input:  You will then return
the clean version of the {tgt}
sentence as output.
```

For each task {src} refers to the source language (English or French) and {tgt} refers to the target language (English, French, or Japanese).

## C Requests

This section shows the requests we used in the prompts. The request for the monolingual cleaning method and generative translation methods are the same and are as follows: `This is the input {input_sent}, .  Please return the desired output in the correct format`. The only difference is that the {input_sent} refers to the target sentence in the case of the monolingual cleaning method, whereas the {input_sent} refers to the source sentence for the generative translation method. The request for the bilingual cleaning task is different because of the multiple inputs. The request for this method is as follows: `These are the inputs {src_sent}, {tgt_sent}.  Please return the desired output in the correct format`.

## D Samples

In Table 4, we show several samples from MTNT, and show how each distinct method tends to clean these samples in different ways. In some samples abbreviations are corrected, in others emojis are removed and some are changed based on language or word choice.

## E Detailed Similarity Scores

For the convenience of the latter works that plan to use our method as a baseline, we list the detailed similarity scores from Figure 2 in Table 5, 6 and 7.

| Eval. Set | LASER | BLEU | Jaccard | Rouge-1 | Jaro Winkler |
|---|---|---|---|---|---|
| Bilingual | 0.94 | 0.90 | 0.79 | 0.66 | 0.54 |
| Translation | 0.89 | 0.81 | 0.61 | 0.44 | 0.26 |
| Monolingual | 0.95 | 0.91 | 0.82 | 0.71 | 0.60 |
| Correction-tool | 0.99 | 0.98 | 0.99 | 0.97 | 0.94 |

Table 5: Similarity scores between noisy and cleaned French-to-English target samples.

| Eval. Set | LASER | BLEU | Jaccard | Rouge-1 | Jaro Winkler |
|---|---|---|---|---|---|
| Bilingual | 0.93 | 0.88 | 0.69 | 0.54 | 0.39 |
| Translation | 0.89 | 0.79 | 0.54 | 0.37 | 0.23 |
| Monolingual | 0.97 | 0.93 | 0.88 | 0.80 | 0.73 |
| Correction-tool | 0.98 | 0.97 | 0.93 | 0.88 | 0.84 |

Table 6: Similarity scores between noisy and cleaned English-to-French target samples.

| Eval. Set | LASER | BLEU | Jaccard | Rouge-1 | Jaro Winkler |
|---|---|---|---|---|---|
| Bilingual | 0.91 | 0.86 | 0.72 | 0.60 | 0.39 |
| Translation | 0.83 | 0.70 | 0.48 | 0.35 | 0.10 |
| Monolingual | 0.96 | 0.93 | 0.90 | 0.85 | 0.76 |
| Correction-tool | 1.00 | 1.00 | 1.00 | 1.00 | 0.95 |

Table 7: Similarity scores between noisy and cleaned English-to-Japanese target samples.

## F Contrastive Learning

### F.1 Training Loss

The contrastive loss (Chen et al., 2020) is computed based on two source sentences: the original source sentences $x$ and the augmented source sentences $z$.

$$\mathcal{L}_{ctr} = -\sum_{i}^{N} \log \frac{\exp(\text{sim}(\boldsymbol{e}_{x^i}, \boldsymbol{e}_{z^i})/\tau)}{\sum_{j}^{N} \exp(\text{sim}(\boldsymbol{e}_{x^i}, \boldsymbol{e}_{z^j})/\tau)}$$

where $x^i$ is the original sentence, $z^i$ is the augmented version of $x^i$, and $\tau$ is the temperature factor. Therefore, the positive sample is the corresponding augmented sentence, while the negative samples are the augmented versions of other original source sentences from the same mini-batch.

$e_{x^i}$ and $e_{z^i}$ are the average representations along the sequence dimension from the encoder outputs.

Apart from the contrastive loss, the standard cross-entropy loss is calculated as:

$$\mathcal{L}_{ce} = -\sum_{i=1}^{N} (\log P_\theta(y^i|x^i) + \log P_\theta(y^i|z^i))$$

We combine both losses as the final loss:

$$\mathcal{L} = \mathcal{L}_{ce} + \lambda \mathcal{L}_{ctr}$$

where $\lambda$ is an interpolation factor. We incorporate the augmented source inputs $z$ to ensure that the model can still generate correct translations with noisy input.

## F.2 Optimal Hyperparameters

We conduct thorough experiments to choose the optimal hyperparameters for contrastive learning.

**Temperature $\tau$.** This hyperparameter plays a crucial role in adjusting the softmax function used in the contrastive learning framework, thereby affecting the distribution of the similarity scores between augmented and original sentences. By varying the value of $\tau$, we can control the concentration or diffusion of the score distribution. Figure 6 shows the results from the models trained with different augmentation strategies. It is evident that $\tau = 0.1$ uniformly performs the best. So we set $\tau = 0.1$ by default.

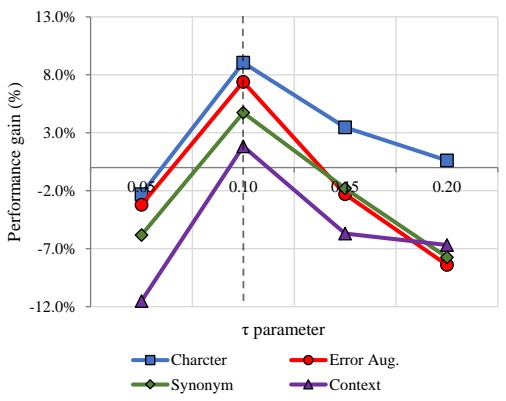

Figure 6: Performance vs. temperature factor $\tau$ for models trained with different augmentation strategies on the French-to-English translation direction. Scores are computed on the bilingual C-MTNT evaluation set.

**Loss balance $\lambda$.** Our final loss consists of the standard cross-entropy loss and the contrastive loss. Here we conduct experiments to choose the optimal loss balance factor $\lambda$. As shown in Figure

7, the optimal $\lambda$ varies for different augmentation methods. We set $\lambda = 0.01$ by default since this value works better for most methods.

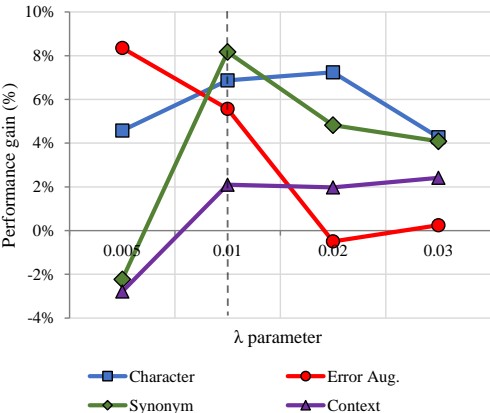

Figure 7: Performance vs. loss balance factor $\lambda$ for models trained with different augmentation strategies on the French-to-English translation direction. Scores are computed on the bilingual C-MTNT evaluation set.

## G Prompt for GPT-4 Evaluation

```
"""In the following, I'm going
to show you one noisy source
sentence in French and one noisy
target sentence in English.  In
addition, I also offer you two
clean versions of the noisy
target sentence.
  Can you rank these two clean
target sentences based on these
three criteria:
  1.  Comprehensive noise removal:
All forms of noise should be
eliminated from the noisy target
sentence, including removing
semantically meaningless emojis,
translating emojis with semantic
content into words, correcting
misspellings, etc.
  2.  Semantic preservation:
The clean target sentence
should retain similar semantic
information as the original noisy
target sentence.
  3.  Alignment with the source
sentence:  The clean target
sentence should convey the
same intended meaning as the
noisy source sentence, ensuring
accurate translation and faithful
```

representation of the original
content.
  Noisy source sentence in French:
{0}
  Noisy target sentence in
English:  {1}
  The first clean target sentence:
{2}
  The second clean target
sentence:  {3}
  For your output, you don't need
to give any explanation.  If
the first version is better, you
output 1.  If the second version
is better, you output 2.  If they
are equally clean, you output
3.""".format(source, target,
target_from_clean_method_v1,
target_from_clean_method_v2)

| Method | Lang. | Noisy Target Sample | Clean Target Sample |
|---|---|---|---|
| Bilingual | EN | ":p I don't have many juicy stories to tell right now."
"If in doubt, tinker with the doc."
"Social network =/= reality but when you add news paper to that..." | "I don't have many juicy stories to tell at the moment."
"If in doubt, tinker with the document."
"Social network does not equal reality, but when you add newspapers to that..." |
| | FR | "Je pense que plus l'on reste céto et moins on a envie de ces choses."
"el oh el Lol merci, j'ai le même espoir pour toi aussi compère"
"Je ne sais pas quoi faire Passe à autre chose." | "Je pense que plus on reste Keto, moins on a envie de ces choses."
"Mort de rire merci, j'ai le même espoir pour toi aussi, compadre"
"Je ne sais pas quoi faire. Allons de l'avant." |
| | JA | "WreslteManiaの試合のマッチの予測は今年は難しいです。"
"トム・クランシーのレインボー・シックス・スリー。😕"
"んんん、たぶん、職場で見るには少し危険かな。" | "今年のWrestleManiaの試合順は予測しにくいです。"
"トム・クランシーのレインボー・シックス・スリー。"
"んんん、ちょっと危ないかもしれないですね。" |
| Monolingual | EN | ":p I don't have many juicy stories to tell right now."
"If in doubt, tinker with the doc."
"Social network =/= reality but when you add news paper to that..." | "I don't have many juicy stories to tell right now."
"If you are unsure, tinker with the document."
"Social network does not equal reality but when you add a newspaper to that..." |
| | FR | "Je pense que plus l'on reste céto et moins on a envie de ces choses."
"el oh el Lol j'ai le même espoir pour toi aussi compère"
"Je ne sais pas quoi faire Passe à autre chose." | "Je pense que plus on reste Keto et moins on a envie de ces choses."
"Mort de rire merci, j'ai le même espoir pour toi aussi compère"
"Je ne sais pas quoi faire. Passe à autre chose." |
| | JA | "WreslteManiaの試合のマッチの予測は今年は難しいです。"
"トム・クランシーのレインボー・シックス・スリー。😕"
"んんん、たぶん、職場で見るには少し危険かな。" | "今年のWrestleManiaの試合順は予想が難しいです。"
"トム・クランシーのレインボー・シックス・スリー。"
"んんん、たぶん、職場で見るには少し危険かもしれません。" |
| Translation | EN | ":p I don't have many juicy stories to tell right now."
"If in doubt, tinker with the doc."
"Social network =/= reality but when you add news paper to that..." | "I have few juicy stories to tell at the moment."

"When in doubt, tweak the document."
"Social media does not equal reality, but when you add journals to that, on the other hand..." |
| | FR | "Je pense que plus l'on reste céto et moins on a envie de ces choses."
"el oh el Lol j'ai le même espoir pour toi aussi compère"
"Je ne sais pas quoi faire Passe à autre chose." | "Je pense que plus nous restons en mode Keto, moins nous avons envie de ces choses."
"Mort de rire merci, j'ai la même espérance pour toi aussi compadre"
"Je ne sais pas quoi faire. Avancer." |
| | JA | "WreslteManiaの試合のマッチの予測は今年は難しいです。"
"トム・クランシーのレインボー・シックス・スリー。😕"
"んんん、たぶん、職場で見るには少し危険かな。" | "今年のWrestleManiaの試合順は予想が難しいです。"
"トム・クランシーのレインボーシックス3。"
"うーん、多分ちょっと、少しはNSFWっぽいかもしれない" |

Table 4: Several noisy target samples from MTNT and C-MTNT with different cleaning methods. The red text is noise, the blue text indicates rephrased parts, and the green text indicates the removal or correction of noise.