# OpenReview forum: "Ask Language Model to Clean Your Noisy Translation Data"
_EMNLP/2023/Conference — EMNLP 2023 Findings_

### Official Review · Reviewer_6pw7 · 2023-07-30

**Typos Grammar Style And Presentation Improvements:** None to my notice
**Soundness:** 3

**Excitement:**

3: Ambivalent: It has merits (e.g., it reports state-of-the-art results, the idea is nice), but there are key weaknesses (e.g., it describes incremental work), and it can significantly benefit from another round of revision. However, I won't object to accepting it if my co-reviewers champion it.

**Missing References:**

I feel these citations for these works should also be present in the final work, but this is up to author's discretion:
1) https://arxiv.org/abs/2302.01398
2) https://aclanthology.org/2022.emnlp-main.649/
3) https://aclanthology.org/2023.findings-acl.381.pdf
4) https://arxiv.org/pdf/2210.00193.pdf

**Paper Topic And Main Contributions:**

Machine Translation approaches and their evaluations may suffer from noisy source as well as target sentences thereby limiting their performances. The paper addresses this challenge by leveraging a large language model (GPT-3.5) to clean the target side of the MTNT dataset and they present a cleaner version called C-MTNT which could be a good resource for the MT community based on authors' claims. The paper also demonstrates the few shot capabilities of large language models and how they can effectively be leveraged for generating parallel corpora tailored to MT use cases for a robust evaluation.

**Questions For The Authors:**

A) While this is not a decision changing question, I wanted to understand why the authors felt that changing emojis to a more meaningful representation is required (mentioned a few times: line 018, 094 etc.). Would it not hurt metric like BLUE when evaluating performance?

B) I would love to hear from authors about their views on my criticism and hear their counterpoints so that I can reconsider my points allocation.

**Reasons To Accept:**

The paper demonstrates the effectiveness of large language models in few-shot machine translation tasks and designs experiments to demonstrate the cases where they can be particularly useful (they show sometimes monolingual C-MTNT does not work as good as expected and without controls LLMs tend to hallucinate sometimes which is a known issue)

One more job that the paper does well is in setting up experiments with language model training to demonstrate the effectiveness of C-MTNT in evaluating model's robustness for machine translations tasks. Instead of directly comparing on BLUE score, authors use relative performance gain in BLEU score which to me seemed like a fairer evaluation metric overall.

Overall, the paper seems to be well written for quick understanding of what the authors are trying to achieve.

**Reasons To Reject:**

While the paper has its merits there are many significant reasons for rejections. The authors' three approaches that they claim as experiments (line 112 to 116) are nothing but few shot NMT capabilities of large language models and hence while designing a modified version of C-MTNT does seem a fair step, I am not sure where the contribution of the paper lies. Also, the paper misses how sensitive the cleaning experiments could be based on what prompting strategies that they follow and how much hallucinations these models suffer from if the focus was just to show the effectiveness of few shot capabilities for correction.

The baseline of using language tool python seems way too easy for large language models to beat, rather, an effective baseline to beat or just evaluate would have been a fair human correction/ evaluation.

The authors use single BLUE metric for their evaluations. While they do show improvements on BLUE based metrics, it would have been interesting to see how the performance corresponded to LM based evaluations metrics like BLEURT-20, COMET-QE.

Also, authors could make further improvements in their work by having leveraged the latest research on parallel corpus that already make certain perturbations in addition to MTNT and give a fair comparison on how their approach is able to perform on different augmentations in addition to the ones that they have used (synonyms, character augmentations). Some of the recent work that I got to read was (https://aclanthology.org/2023.findings-acl.381.pdf, https://aclanthology.org/2022.emnlp-main.649.pdf etc.) There is a body of multiple similar work that authors can take a look at and evaluate how they can leverage for improving their experiments.

Authors might also want to cite some of the works in field which I found was missing.

**Reproducibility:**

4: Could mostly reproduce the results, but there may be some variation because of sample variance or minor variations in their interpretation of the protocol or method.

**Reviewer Confidence:**

4: Quite sure. I tried to check the important points carefully. It's unlikely, though conceivable, that I missed something that should affect my ratings.

---

> ### Author Rebuttal · Authors · 2023-08-29
>
> Dear reviewer, we would like to thank you for your time, effort, and thorough reviews! We are really encouraged by the reviews that highlight our work:
>
> * Design suitable application case where few-shot MT can be particularly useful;
> * Offers well-set-up experiments to demonstrate the effectiveness of collected data, C-MTNT;
> * Is well written for quick understanding.
>
> We have responded to your comments below and updated our draft accordingly.
>
> If you still have additional questions and suggestions, we would be happy to answer your questions and incorporate the suggestions into an updated draft.
>
> ---
>
> > Reasons To Reject: where the contribution of the paper lies.
>
> We acknowledge that our work builds upon established techniques, but we'd like to highlight the novelty and advantages of our approach:
>
> * **Application novelty**: We deem both methodological and application novelty equally important. Compared to previous work [1] where few-shot learning is applied to prompt an LLM to obtain a better translation, we apply few-shot learning in building a new benchmark to evaluate a MT system, which is normally done by human annotators. We understand that some previous works, like Alpaca, already applied LLM to create a new training set. Compared to them, our job focuses on data cleaning, a more challenging task that requires the understanding of the whole sentence while still being able to detect noise and offer correction. We didn't find any related work in this specific area with LLM (We're glad for your references), since data cleaning normally requires extensive work of human annotators.
>
> * **New way to construct prompt examples**: We are really sorry for forgetting to include this novel part in the paper. Compared to chain-of-thought where human annotators are asked to offer some representative few-shot examples, the prompt examples in our paper are generated by GPT-3.5 itself. If you are interested in how we prompt GPT-3.5 to generate these examples, we kindly ask you to go to our response to Reviewer pKJv on "Question 3. How are the prompt examples selected and how sensitive is the LLM to the choice of these examples?". This part was already added to our updated draft.
>
> * **New benchmark**: Our original goal of this research was to evaluate the robustness of different MT systems against noisy input. However, the most suitable benchmark we could find is MTNT where noise is almost equally distributed in both source and target sides, which is not desired for real-life scenarios, like e-commerce. By designing three cleaning methods, we offer a new benchmark to evaluate the robustness of MT systems. In addition, these three methods are also suitable for different scenarios. When only source sentences are available, translation cleaning is a better choice. When both source and target sentences are available, bilingual cleaning is a better choice.
>
> * **Thorough evaluation of the new benchmark**: In the paper, we offer multiple ways to evaluate the effectiveness of the newly constructed benchmark: (1) Measure the semantic similarity between noisy and cleaned target sentences in Figure 2; (2) Quantifiably measure the amount of different types of noise in the benchmark in Table 2; (3) Conduct MT experiments to show the effectiveness of the newly constructed benchmark; (4) Thanks to your advice, we also include human and GPT-4 evaluation. You can find more details in the following question.
>
> ---
>
> > Reasons To Reject: The baseline of using language tool python seems way too easy for large language models to beat, rather, an effective baseline to beat or just evaluate would have been a fair human correction/ evaluation.
>
> Traditional cleaning method serves more as a threshold in our work instead of a strong baseline. Similar to Table 2 where we include the noise distribution in clean benchmarks, we want to show a lower bound for the reader's easy comparison. Self-comparison among different cleaning methods, like monolingual, bilingual, and translation, are our focus.
>
> Thanks to your advice, we conducted an emergency human and GPT-4 evaluation experiment on C-MTNT. For your convenience, we first show the conclusion here: **Overall, human and GPT-4 evaluation share a similar preference: bilingual > translation > monolingual, which is also similar to our originally drawn conclusion from Table 3 in the paper. Compared to GPT-4, human annotators prefer to vote for "Tie". We argue the main reason is that most cleaned sentences are very similar with a low level of noise, which further justifies the effectiveness of our proposed methods.**
>
>
> The experimental setting is:
>
> (1) 100 sentences from C-MTNT Fr->En are sampled to generate three files. These three files are about the binary comparison of bilingual vs. monolingual, bilingual vs. translation, and monolingual vs. translation. The order of sentences, including their indexes and which cleaning method comes first, is randomly shuffled. There is no chance for the annotator to guess which sentence corresponds to which method, and which file corresponds to the comparison between which two methods. (Notably, we prefer binary comparison to ranking over three methods, since it's easier for human annotators. In addition, the cleaned sentences from the correction tool are excluded, since they are too easy to be beaten.)
>
> (2) Four annotators (authors of this paper) are asked to give their preferences for each comparison, based on our three criteria of comprehensive noise removal, semantic preservation, and alignment with the source sentence (Lines 90-103). If both sentences show a similar level of noise, they are asked to give a "Tie".
>
> (3) Due to the time limit, we couldn't compare all sentences from different translation directions, and hire a larger amount of annotators. To overcome this limit, we also prompt GPT-4 to do the comparison on all sentences on the Fr<->En corpora. En->Ja is excluded since GPT-4 is not good at Japanese, as stated in our paper (Line 314-322). We append the prompt at the end of this response, i.e. Appendix A.
>
> Here is the result from human evaluation on the sampled 100 sentences of Fr->En.
>
> |  | Left is better | Tie | Right is better | |
> | --- | :---: | :---: | :---: | --- |
> | Bilingual | **30.0%** | 46.8% | 23.2% | Monolingual |
> | Bilingual | **32.2%** | 38.0% | 29.8% | Translation |
> | Monolingual | 27.0% | 39.5% | **33.5%** | Translation |
>
> The GPT-4 evaluation result of these 100 samples is:
> |  | Left is better | Tie | Right is better | |
> | --- | :---: | :---: | :---: | --- |
> | Bilingual | **34.0%** | 37.0% | 29.0% | Monolingual |
> | Bilingual | **40.0%** | 26.0% | 34.0% | Translation |
> | Monolingual | 35.0% | 21.0% | **44.0%** | Translation |
>
> Overall, human and GPT-4 evaluations share a similar preference: bilingual > translation > monolingual. Compared to GPT-4, human annotators prefer to vote for "Tie". We argue the main reason is that most cleaned sentences are very similar with a low level of noise, which further justifies the effectiveness of our proposed methods.
>
> The results from the GPT-4 evaluation on the whole C-MTNT are:
>
> En -> Fr
>
> |  | Left is better | Tie | Right is better | |
> | --- | :---: | :---: | :---: | --- |
> | Bilingual | **42.3%** | 32.0% | 25.7% | Monolingual |
> | Bilingual | **45.8%** | 21.2% | 33.0% | Translation |
> | Monolingual | 31.3% | 16.8% | **51.9%** | Translation |
>
> Fr -> En
> |  | Left is better | Tie | Right is better | |
> | --- | :---: | :---: | :---: | --- |
> | Bilingual | **35.5%** | 35.7% | 28.8% | Monolingual |
> | Bilingual | **40.2%** | 22.8% | 37.0% | Translation |
> | Monolingual | 35.4% | 15.3% | **49.3%** | Translation |
>
> The conclusion is similar to the above discussion, i.e. bilingual > translation > monolingual, which is also similar to our originally drawn conclusion from Table 3 in the paper. We have added these new results to our updated draft. And the newly collected evaluation scores will also be released later.
>
> ---
>
> > Reasons To Reject: the paper misses how sensitive the cleaning experiments could be based on what prompting strategies that they follow and how much hallucinations these models suffer from if the focus was just to show the effectiveness of few shot capabilities for correction.
>
> We agree with this point since they are well-known issues for LLMs. We conduct some experiments to answer your concerns.
>
> 1. **The sensitivity of different prompting strategies**
>
> We conduct new experiments by designing new prompt examples. Similar to the human evaluation part, we only collect 100 clean sentences on Fr->En with the new prompt examples, keeping the other parts of Figure 1 the same for the ablation study. After collection, we use the same prompt in Appendix A to ask GPT-4 to give its preference, since it shares a similar preference as human annotators.
>
> First, we design two new sets of prompt examples for the bilingual cleaning method to check whether new prompt examples cause large variances for the same cleaning method. The results are:
> | | Left is better | Tie | Right is better |  |
> | --- | --- |--- |--- |--- |
> | Original prompt examples | 18% | 60% | 12% | New prompt examples v1 |
> | Original prompt examples | 20% | 53% | 27% | New prompt examples v2 |
>
> We do see some variance with different prompt examples. However, the major preference (>50%) of GPT-4 is located in "Tie", indicating that the same cleaning method with different prompt examples offers similar cleaned sentences.
>
> Second, we design two more sets of prompt examples for the monolingual and translation cleaning methods to check whether we have a similar pattern as the previous human/GPT-4 evaluation. The results are:
> | | Left is better | Tie | Right is better |  |
> | --- | --- |--- |--- |--- |
> | Original bilingual | **36%** | 40% | 24% | Monolingual with new prompt examples |
> | Original bilingual | **34%** | 35% | 31% | Translation with new prompt examples |
> | Monolingual with new prompt examples| 29% | 31% | **40%** | Translation with new prompt examples |
>
> A similar pattern, i.e. bilingual > translation > monolingual, is observed. **In sum, different prompt examples will cause a mild variance, but the observed pattern is similar.**
>
>
> 2. **How many hallucinations do these models suffer from**
>
> We did see some hallucinations from the cleaned sentences. It happens more frequently for the monolingual cleaning method since there is no supervision from the target side. One trick we applied to reduce this issue is including the LASER score (Lines 258-264). If the LASER score between the cleaned and noisy sentences is too low (<0.7), we repeat the API call till it is larger than 0.7. In this way, we offer some room for cleaning while still requiring a certain degree of similarity between the cleaned and noisy sentences to avoid hallucinations. We are open to other suggestions.
>
> ---
>
> > Reasons To Reject: The authors use a single BLUE metric for their evaluations. While they do show improvements on BLUE based metrics, it would have been interesting to see how the performance corresponded to LM based evaluations metrics like BLEURT-20, COMET-QE.
>
> We totally agree with this point, since LM-based metrics show more correlation with human preference. Apart from these LM-based metrics, we also offer GPT-4 evaluation in the above question, which shows a similar finding as the BLEU metric. The results of other LM-based metrics will be included in our next updated version.
>
> ---
>
> > Reasons To Reject: authors could make further improvements in their work by having leveraged the latest research on parallel corpus that already make certain perturbations in addition to MTNT and give a fair comparison on how their approach is able to perform on different augmentations in addition to the ones that they have used. There is a body of multiple similar work that authors can take a look at and evaluate how they can leverage for improving their experiments.
>
> This is a very inspired suggestion. We do find two of your offered references [2, 3] quite related. The implementation/discussion/comparison of these two works is on our to-do list.
>
> ---
>
> > Reasons To Reject: Authors might also want to cite some of the works in the field which I found was missing.
>
> Really thank you for the offered references [1, 2, 3, 4]. We find all of them very related and already included them in our related work.
>
> ---
>
> > Question A. why the authors felt that changing emojis to a more meaningful representation is required (mentioned a few times: line 018, 094 etc.). Would it not hurt metric like BLUE when evaluating performance?
>
> This is a very good question. Whether emojis are noise depends on the application case (similar to slang). For the social media use case, we think emojis are not necessarily a noise. However, for some official use cases, like the product title in e-commerce, emojis are considered as noise. In our work, we prefer to treat emojis as noise, because (1) It makes our task more challenging, which could explore the potential of LLM; (2) We want to expand the applicable case of our benchmark, making it a suitable benchmark for different MT systems.
>
> We believe changing emojis to some meaningful representation would hurt the metric, like BLEU, because it is easier for a MT model to directly copy the emoji from the source sentence to the target sentence (emojis are normally the same on both sides) than to translate the emojis to some words according to different contexts. This is also one reason why we care more for the relative improvement rather than the absolute improvement in Section 4.5. However, we still think it is reasonable for such changing: (1) Making our benchmark more challenging; (2) Making our benchmark more applicable for different MT systems.
>
> ---
>
> > Missing References: I feel these citations for these works should also be present in the final work.
>
> Thank you for these references, we already added them [1, 2, 3, 4] to our related work in an updated draft.
>
> ---
>
> References
>
> [1] Garcia, X., Bansal, Y., Cherry, C., Foster, G., Krikun, M., Johnson, M., & Firat, O. (2023, July). The unreasonable effectiveness of few-shot learning for machine translation.
>
> [2] Karpinska, M., Raj, N., Thai, K., Song, Y., Gupta, A., & Iyyer, M. (2022). DEMETR: Diagnosing evaluation metrics for translation.
>
> [3] Jwalapuram, P. (2023, July). Pulling Out All The Full Stops: Punctuation Sensitivity in Neural Machine Translation and Evaluation.
>
> [4] Riley, P., Dozat, T., Botha, J. A., Garcia, X., Garrette, D., Riesa, J., ... & Constant, N. (2022). Frmt: A benchmark for few-shot region-aware machine translation.
>
> ---
> Appendix A: Prompt for GPT-4 evaluation
> ```
> def generate_prompt(source, target, v1, v2):
>     return """In the following, I'm going to show you one noisy source sentence in French and one noisy target sentence in English. In addition, I also offer you two clean versions of the noisy target sentence.
>
> Can you rank these two clean target sentences based on these three criteria:
> 1. Comprehensive noise removal: All form of noise should be eliminated from the noisy target sentence, including removing semantically meaningless emojis, translating emojis with semantic content into words, correcting misspellings, etc.
> 2. Semantic preservation: The clean target sentence should retain similar semantic information as the original noisy target sentence.
> 3. Alignment with the source sentence: The clean target sentence should convey the same intended meaning as the noisy source sentence, ensuring accurate translation and faithful representation of the original content.
>
> Noisy source sentence in French: {0}
> Noisy target sentence in English: {1}
> The first clean target sentence: {2}
> The second clean target sentence: {3}
>
> For your output, you don't need to give any explanation. If the first version is better, you output 1. If the second version is better, you output 2. If they are equally clean, you output 3.""".format(source, target, v1, v2)
> ```

---

### Official Review · Reviewer_Z2kB · 2023-08-03

**Soundness:** 3

**Excitement:**

2: Mediocre: This paper makes marginal contributions (vs non-contemporaneous work), so I would rather not see it in the conference.

**Paper Topic And Main Contributions:**

Neural machine translation models exhibit a noticeable decline in translation quality when exposed to noisy input. The MTNT dataset (Michel and Neubig, 2018) is widely used as a benchmark for evaluating the robustness of NMT models against noisy input. This paper argues that the noise exists not only in the source input but also in the target output, which affects the use of this dataset. Therefore, this paper proposes to clean the noise from the target sentences in MTNT by leveraging the large language models.

**Questions For The Authors:**

Why did not introduce the request in lines 260-261  in the first place?   What is the maximum number of iterations?

**Reasons To Accept:**

A cleaned datatset is helpful for exploring the application of machine translation in real scenarios.

**Reasons To Reject:**

The paper should be strengthened in the writting, which is not fully clear now.  For contrastive learning, which is the positive sample? Which is the negative sample? Please explain in detail in Section 4.4. Please provide a more detailed description on cleaning data by leveraging the large language models.
It is hard to understand the conclusion in lines 505-515. It is necessary to introduce human evaluation to verify that the new dataset is better.
For the baseline, why did you just use language_tool_python module? Why didn't you use the emoji library, better_profanity, and profanity_check?



**Reproducibility:**

4: Could mostly reproduce the results, but there may be some variation because of sample variance or minor variations in their interpretation of the protocol or method.

**Reviewer Confidence:**

3: Pretty sure, but there's a chance I missed something. Although I have a good feel for this area in general, I did not carefully check the paper's details, e.g., the math, experimental design, or novelty.

**Typos Grammar Style And Presentation Improvements:**

The text-davinci-003 model is the instructGPT model, which introduces SFT and RLHF.

---

> ### Author Rebuttal · Authors · 2023-08-29
>
> Dear reviewer, we would like to thank you for your time, effort, and thorough reviews! We are really encouraged by the reviews that highlight our work:
>
> * Offers a helpful and cleaned dataset for exploring the application of MT models in real-life scenarios.
>
> We have responded to your comments below and updated our draft accordingly.
>
> If you still have additional questions and suggestions, we would be happy to answer your questions and incorporate the suggestions into an updated draft.
>
> ---
>
> > Reasons To Reject: The paper should be strengthened in the writting, which is not fully clear now. For contrastive learning, which is the positive sample? Which is the negative sample? Please explain in detail in Section 4.4.
>
> Sorry for the unclear explanation of contrastive learning. You can find more details in Appendix F.1. For your convenience, we show the related contents here. We calculated the contrastive loss as:
>
> numerator = $exp(sim(e_{x^i}, e_{z^i}))/\tau$
>
> denominator = $\sum_j^Nexp(sim(e_{x^i}, e_{z^j})/\tau$
>
> $\mathcal{L}_{ctr}$  = - $\sum_i^{N}$log$\frac{numerator}{denominator}$
>
> where $x^i$ is the original source sentence, $z^i$ is the augmented version of $x^i$. Therefore, **the positive sample is the corresponding augmented sentence, while the negative samples are the augmented versions of other original source sentences from the same mini-batch.**
>
> We have made a modification to Section 4.4 in our updated draft for a clear explanation.
>
> ---
>
> > Reasons To Reject: Please provide a more detailed description on cleaning data by leveraging the large language models.
>
> Thank you for raising the question regarding the details of how we use LLM for data cleaning in our work. We recognize the need for more explicit elucidation and will add further clarifications in the revised paper.
>
> 1. **Method variants**: In our approach, we have three cleaning methods tailored to different data scenarios: Bilingual, Monolingual, and Translation. These methods leverage the capabilities of the LLM to produce clean text that preserves the essential information in noisy data.
>
> * **Bilingual**: For bilingual cleaning, we input both the noisy source and target samples. The aim is to generate a clean target text that maintains its alignment with the noisy source.
> * **Monolingual**: For this approach, we only provide a noisy target text and the LLM generates a cleaner version of it.
> * **Translation**: This technique utilizes the LLM to translate a noisy source text into a clean target text, effectively ignoring the source’s noise.
>
> 2. **Chain-of-thought prompting**: We leverage recent advancements in prompting methods to improve our LLM’s understanding of what we aim to achieve. Through chain-of-thought prompting, we can guide the model’s behavior and optimize its responses.
>
> 3. **Prompt design**: Our prompts contain multiple components, including the description of the context, method formulation, desired output framework, few-shot examples, and then the input samples. This composite prompt design improves the cleaning operation by making it more aligned with our objectives.
>
> 4. **Semantic similarity**: To ensure that the cleaned text retains the intended meaning of the original text, we compute semantic similarity scores using LASER. If the similarity score is below a certain threshold, we iterate the cleaning process with additional guidance to the model.
>
> 5. **Feedback loop with LASER**: In cases where the semantic similarity score falls below our threshold (0.7), we modify the prompt to include additional guidance, ensuring the cleaned text better aligns with the original meaning. We iterate this process until the threshold is met.
>
> In summary, our approach leverages both the versatility of LLMs and semantic similarity checks to achieve effective and meaningful data cleaning. Additional computational details and algorithmic procedures will be incorporated in the revision to provide a thorough understanding of our data-cleaning process.
>
> ---
>
> > Reasons To Reject: It is hard to understand the conclusion in lines 505-515.
>
> The logic behind the conclusion in Lines 505-515 is not straightforward. We offer a more clear explanation here.
>
> Our goal in Section 4.5 is to show which test set is more suitable as a noise evaluation dataset. However, it's not easy to compare different test sets, since they have different sentences. For simplicity, we take the original noisy MTNT test set (termed as MTNT) and MTNT cleaned by bilingual method (termed as C-MTNT) as examples. MTNT and C-MTNT share the same noisy source sentences, but C-MTNT has a cleaner version of target sentences. If a model obtains a better BLUE score on MTNT than on C-MTNT, can we say that the model performs better on MTNT? No, we can't say that, because the target sentences of MTNT and C-MTNT are different. After cleaning, their target sentence length, noise level, and so on are different.
>
> Therefore, we focus on the relative improvement $G$ rather than the absolute improvement. If a model originally performs worse on C-MTNT than on MTNT, and achieves a better BLEU score on C-MTNT after augmented training, we can say that the augmented training offers more gain to C-MTNT. It also demonstrates that the distribution of the augmented dataset is more similar to C-MTNT, so the model has more gain on C-MTNT.
>
> Let's review how we augment our dataset. We only input noise, like character augmentation, to the source sentence while keeping the target sentence clean. This is the key motivation for the whole paper. I.e. we want a model that can generate clean target sentences from noisy source sentences. If a model trained on this augmented dataset achieves a better relative improvement on a test set, we can presume that this test set and the augmented dataset have similar distributions. I.e. most noise is located in the source side rather than in the target side. And this test set is exactly what we want.
>
> According to Figure 4, C-MTNT by bilingual or translation cleaning is our final ideal test set, since the model trained on the augmented dataset offers the highest relative improvement on them. We have added a more clear explanation to this conclusion in our updated draft.
>
> ---
>
> > Reasons To Reject: It is necessary to introduce human evaluation to verify that the new dataset is better.
>
> We totally agree with this point.
>
> In this rebuttal week, we conducted an emergency human and GPT-4 evaluation experiment on C-MTNT. For your convenience, we first show the conclusion here: **Overall, human and GPT-4 evaluation share a similar preference: bilingual > translation > monolingual, which is also similar to our originally drawn conclusion from Table 3 in the paper. Compared to GPT-4, human annotators prefer to vote for "Tie". We argue the main reason is that most cleaned sentences are very similar with a low level of noise, which further justifies the effectiveness of our proposed methods.**
>
> The experimental setting is:
>
> (1) 100 sentences from C-MTNT Fr->En are sampled to generate three files. These three files are about a binary comparison of bilingual vs. monolingual, bilingual vs. translation, and monolingual vs. translation. The order of sentences, including their indexes and which cleaning method comes first, is randomly shuffled. There is no chance for the annotator to guess which sentence corresponds to which method, and which file corresponds to the comparison between which two methods. (Notably, we prefer binary comparison to ranking over three methods, since it's easier for human annotators. In addition, the cleaned sentences from the correction tool are excluded, since they are too easy to be beaten.)
>
> (2) Four annotators (authors of this paper) are asked to give their preferences for each comparison, based on our three criteria of comprehensive noise removal, semantic preservation, and alignment with the source sentence (Lines 90-103). If both sentences show a similar level of noise, they are asked to give a "Tie".
>
> (3) Due to the time limit, we couldn't compare all sentences from different translation directions and hired a larger amount of annotators. To overcome this limit, we also prompt GPT-4 to do the comparison on all sentences on the Fr<->En corpora. En->Ja is excluded since GPT-4 is not good at Japanese, as stated in our paper (Line 314-322). We append the prompt at the end of this response, i.e. Appendix A.
>
> Here is the result from human evaluation on the sampled 100 sentences of Fr->En.
>
> |  | Left is better | Tie | Right is better | |
> | --- | :---: | :---: |:---:| --- |
> | Bilingual | **30.0%** | 46.8% | 23.2% | Monolingual |
> | Bilingual | **32.2%** | 38.0% | 29.8% | Translation |
> | Monolingual | 27.0% | 39.5% | **33.5%** | Translation |
>
> The GPT-4 evaluation result of these 100 samples is:
> |  | Left is better | Tie | Right is better | |
> | --- | :---: | :---: |:---:| --- |
> | Bilingual | **34.0%** | 37.0% | 29.0% | Monolingual |
> | Bilingual | **40.0%** | 26.0% | 34.0% | Translation |
> | Monolingual | 35.0% | 21.0% | **44.0%** | Translation |
>
> Overall, human and GPT-4 evaluations share a similar preference: bilingual > translation > monolingual. Compared to GPT-4, human annotators prefer to vote for "Tie". We argue the main reason is that most cleaned sentences are very similar with a low level of noise, which further justifies the effectiveness of our proposed methods.
>
> The results from the GPT-4 evaluation on the whole C-MTNT are:
>
> En -> Fr
>
> |  | Left is better | Tie | Right is better | |
> | --- | :---: | :---: |:---:| --- |
> | Bilingual | **42.3%** | 32.0% | 25.7% | Monolingual |
> | Bilingual | **45.8%** | 21.2% | 33.0% | Translation |
> | Monolingual | 31.3% | 16.8% | **51.9%** | Translation |
>
> Fr -> En
> |  | Left is better | Tie | Right is better | |
> | --- | :---: | :---: |:---:| --- |
> | Bilingual | **35.5%** | 35.7% | 28.8% | Monolingual |
> | Bilingual | **40.2%** | 22.8% | 37.0% | Translation |
> | Monolingual | 35.4% | 15.3% | **49.3%** | Translation |
>
> The conclusion is similar to the above discussion, i.e. bilingual > translation > monolingual, which is also similar to our originally drawn conclusion from Table 3 in the paper. We have added these new results to our updated draft. And the newly collected evaluation scores will also be released later.
>
> ---
>
> > Reasons To Reject: For the baseline, why did you just use the language_tool_python module? Why didn't you use the emoji library, better_profanity, and profanity_check?
>
> For our baseline, we prioritized tools that not only identify errors but also provide corrections. The language_tool_python module offers this feature.
>
> Using better_profanity, profanity_check, or simply removing profanities can significantly alter the sentence's meaning, which deviates from our criteria of semantic preservation and alignment with the source sentence (Lines 96-103).
>
> Similarly for the emoji, the emoji library can only detect emojis without offering replacement/correction. Simply removing them presupposes they don't add value to the sentence. It is a presumption we didn't want to make, because they do offer meaning to the sentences according to our observation, mostly strengthening the emotion.
>
> ---
>
> > Questions: Why did not introduce the request in lines 260-261 in the first place? What is the maximum number of iterations?
>
> This is a really good observation. Originally, we prompted GPT-3.5 to generate C-MTNT with different cleaning methods without the request. When we checked the generated sentences, we realized that GPT-3.5  unintentionally hallucinates, especially for monolingual cleaning. We didn't want to waste the clean generation from the first iteration. Therefore, we added this request at the end to re-emphasize our desired output for other iterations.
>
> In addition, another benefit of adding the request afterward is to minimize the chances of receiving repetitive responses from the model. While it's not a guarantee that the model will always produce the same answer, the modification helps in reducing redundancy. It's reminiscent of the saying, "The definition of insanity is doing the same thing and expecting different results."
>
> Though there were no cases of the model exceeding this limit, we set **a limit of 10 tries (iterations)** per sentence to avoid excessive API calls.
>
> ---
>
> > Presentation Improvements: The text-davinci-003 model is the instructGPT model, which introduces SFT and RLHF.
>
> Thank you for this correction, we already made an update to our draft.
>
> ---
> Appendix A: Prompt for GPT-4 evaluation
> ```
> def generate_prompt(source, target, v1, v2):
>     return """In the following, I'm going to show you one noisy source sentence in French and one noisy target sentence in English. In addition, I also offer you two clean versions of the noisy target sentence.
>
> Can you rank these two clean target sentences based on these three criteria:
> 1. Comprehensive noise removal: All forms of noise should be eliminated from the noisy target sentence, including removing semantically meaningless emojis, translating emojis with semantic content into words, correcting misspellings, etc.
> 2. Semantic preservation: The clean target sentence should retain similar semantic information as the original noisy target sentence.
> 3. Alignment with the source sentence: The clean target sentence should convey the same intended meaning as the noisy source sentence, ensuring accurate translation and faithful representation of the original content.
>
> Noisy source sentence in French: {0}
> Noisy target sentence in English: {1}
> The first clean target sentence: {2}
> The second clean target sentence: {3}
>
> For your output, you don't need to give any explanation. If the first version is better, you output 1. If the second version is better, you output 2. If they are equally clean, you output 3.""".format(source, target, v1, v2)

---

### Official Review · Reviewer_pKJv · 2023-08-05

**Typos Grammar Style And Presentation Improvements:** Table 3 numbers are a bit too small t…
**Soundness:** 3

**Excitement:**

3: Ambivalent: It has merits (e.g., it reports state-of-the-art results, the idea is nice), but there are key weaknesses (e.g., it describes incremental work), and it can significantly benefit from another round of revision. However, I won't object to accepting it if my co-reviewers champion it.

**Paper Topic And Main Contributions:**

This paper investigates the possibility of using LLMs to clean up the references of MTNT, a noisy MT dataset on both the source and the reference sides. It studies three ways for prompting LLMs to clean up the reference by 1) looking only at the target; 2) looking at both the target and the source; 3) looking at the source only (i.e., translation). The analysis shows that the second approach actually delivers the best result in terms of preserving the original meaning and reducing the noise. Through experiments on data augmentation, the paper also argues  that the cleaned dataset is better at evaluating the robustness of MT systems.

**Questions For The Authors:**

1. In Table 2, why is the numbers in the spelling error column not 0 for the correction-tool rows? Does it mean that the tool can recognize more errors than correct them?
2. Would the collected slangs (line 295-297) be released?
3. How are the prompt examples selected and how sensitive is the LLM to the choice of these examples?

**Reasons To Accept:**

1. I find the idea of using LLMs to clean up references quite interesting, and it addresses the issue of insufficient coverage of the traditional rule-based cleaners.
2. The experimental results on three different ways of cleaning the references are insightful and very helpful for future research.

**Reasons To Reject:**

I personally find the first half of the paper (i.e., on cleaning up the data) more convincing than the second half (i.e., on showcasing C-MTNT's effectiveness to evaluate MT robustness). I concede the point that it's nontrivial to say which test set is more suitable as a benchmark since we are comparing different test sets, but I don't find G (introduced in eq 2) a satisfactory solution -- would a 20->22 increase in BLEU actually be better than a 30->32 increase? We would conclude 'yes' if we use G, but I'm not entirely sure. I would have been happier with some sort of human evaluation/auditing of C-MTNT than the data augmentation experiments that make up the latter half of the paper.

**Reproducibility:**

3: Could reproduce the results with some difficulty. The settings of parameters are underspecified or subjectively determined; the training/evaluation data are not widely available.

**Reviewer Confidence:**

3: Pretty sure, but there's a chance I missed something. Although I have a good feel for this area in general, I did not carefully check the paper's details, e.g., the math, experimental design, or novelty.

---

> ### Author Rebuttal · Authors · 2023-08-28
>
> Dear reviewer, we would like to thank you for your time, effort, and thorough reviews! We are really encouraged by the reviews that highlight our work:
>
> * Offers an interesting idea of using LLM to clean up references;
>
> * Addresses the insufficient coverage issue of traditional rule-based cleaning methods;
>
> * Offers insightful and very helpful experimental results for future research.
>
> We have responded to your comments below and updated our draft accordingly.
>
> If you still have additional questions and suggestions, we would be happy to answer your questions and incorporate the suggestions into an updated draft.
>
> ---
>
> > Reasons To Reject: I concede the point that it's nontrivial to say which test set is more suitable as a benchmark since we are comparing different test sets, but I don't find G (introduced in eq 2) a satisfactory solution -- would a 20->22 increase in BLEU actually be better than a 30->32 increase? We would conclude 'yes' if we use G, but I'm not entirely sure. I would have been happier with some sort of human evaluation/auditing of C-MTNT than the data augmentation experiments that make up the latter half of the paper.
>
> We totally agree with the point that 20->22 increment in BLEU isn't necessarily better than 30->32 increment, even though our metric G implies it.
>
> Thanks to your advice, we conducted an emergency human and GPT-4 evaluation experiment on C-MTNT. For your convenience, we first show the conclusion here: **Overall, human and GPT-4 evaluation share a similar preference: bilingual > translation > monolingual, which is also similar to our originally drawn conclusion from Table 3 in the paper. Compared to GPT-4, human annotators prefer to vote for "Tie". We argue the main reason is that most cleaned sentences are very similar with a low level of noise, which further justifies the effectiveness of our proposed methods.**
>
> The experimental setting is:
>
> (1) 100 sentences from C-MTNT Fr->En are sampled to generate three files. These three files are about binary comparisons of bilingual vs. monolingual, bilingual vs. translation, and monolingual vs. translation. The order of sentences, including their indexes and which cleaning method comes first, is randomly shuffled. There is no chance for the annotator to guess which sentence corresponds to which method, and which file corresponds to the comparison between which two methods. (Notably, we prefer binary comparison to ranking over three methods, since it's easier for human annotators. In addition, the cleaned sentences from the correction tool are excluded, since they are too easy to be beaten.)
>
> (2) Four annotators (authors of this paper) are asked to give their preferences for each comparison, based on our three criteria of comprehensive noise removal, semantic preservation, and alignment with the source sentence (Lines 90-103). If both sentences show a similar level of noise, they are asked to give a "Tie".
>
> (3) Due to the time limit, we couldn't compare all sentences from different translation directions, and hire a larger amount of annotators. To overcome this limit, we also prompt GPT-4 to do the comparison on all sentences on the Fr<->En corpora. En->Ja is excluded since GPT-4 is not good at Japanese, as stated in our paper (Line 314-322). We append the prompt at the end of this response, i.e. Appendix A.
>
> Here is the result from human evaluation on the sampled 100 sentences of Fr->En.
>
> |  | Left is better | Tie | Right is better | |
> | --- | :---: | :---: | :---: | --- |
> | Bilingual | **30.0%** | 46.8% | 23.2% | Monolingual |
> | Bilingual | **32.2%** | 38.0% | 29.8% | Translation |
> | Monolingual | 27.0% | 39.5% | **33.5%** | Translation |
>
> The GPT-4 evaluation result of these 100 samples is:
> |  | Left is better | Tie | Right is better | |
> | --- | :---: | :---: | :---: | --- |
> | Bilingual | **34.0%** | 37.0% | 29.0% | Monolingual |
> | Bilingual | **40.0%** | 26.0% | 34.0% | Translation |
> | Monolingual | 35.0% | 21.0% | **44.0%** | Translation |
>
> Overall, human and GPT-4 evaluations share a similar preference: bilingual > translation > monolingual. Compared to GPT-4, human annotators prefer to vote for "Tie". We argue the main reason is that most cleaned sentences are very similar with a low level of noise, which further justifies the effectiveness of our proposed methods.
>
> The results from the GPT-4 evaluation on the whole C-MTNT are:
>
> En -> Fr
>
> |  | Left is better | Tie | Right is better | |
> | --- | :---: | :---: | :---: | --- |
> | Bilingual | **42.3%** | 32.0% | 25.7% | Monolingual |
> | Bilingual | **45.8%** | 21.2% | 33.0% | Translation |
> | Monolingual | 31.3% | 16.8% | **51.9%** | Translation |
>
> Fr -> En
> |  | Left is better | Tie | Right is better | |
> | --- | :---: | :---: | :---: | --- |
> | Bilingual | **35.5%** | 35.7% | 28.8% | Monolingual |
> | Bilingual | **40.2%** | 22.8% | 37.0% | Translation |
> | Monolingual | 35.4% | 15.3% | **49.3%** | Translation |
>
> The conclusion is similar to the above discussion, i.e. bilingual > translation > monolingual, which is also similar to our originally drawn conclusion from Table 3 in the paper. We have added these new results to our updated draft. And the newly collected evaluation scores will also be released later.
>
> ---
>
> > Question 1. In Table 2, why is the numbers in the spelling error column not 0 for the correction-tool rows? Does it mean that the tool can recognize more errors than correct them?
>
> In the process of rectifying grammar and spelling errors, the correction tool occasionally generates sentences that deviate from conventional structures or inadvertently introduce new grammatical or spelling errors. Consequently, upon post-correction verification, a minimal number of discrepancies might still be identified, even though they are significantly reduced compared to the original noisy sentence.
>
> ---
>
> > Question 2. Would the collected slangs (line 295-297) be released?
>
> Due to the time limit, we couldn't release all experimental materials upon submission. However, the release of all materials for reproduction is on our plan. If you still have other requests, please do let us know.
>
> ---
>
> > Question 3. How are the prompt examples selected and how sensitive is the LLM to the choice of these examples?
>
> Thank you for raising this important question regarding the selection of prompt examples and the sensitivity of the LLM to these choices.
>
> 1. **How are the prompt examples selected?**
>
> The prompt examples are initially generated by the model (GPT-3.5) itself with the following prompts:
>
> ```
> Generate 20 examples of parallel sentence pairs where each pair consists of a noisy source sentence in {source_language} and its noisy {target_language} translation. The noise could be in the form of emojis, slang, jargon, slurs, grammatical errors, spelling errors, or code-switching. Also, provide the cleaned-up version of the {target_language} translation for each example. Thus a version of {target_language} translation without any of the aforementioned noise types.
> ```
>
> after which they are subjected to a manual selection process to guarantee their quality, relevance, and efficacy. Our selection process consists of several key steps:
>
> * **Relevance to objective**: Each generated example is first evaluated against the primary objectives of the paper or study. This ensures that the examples serve as accurate representations of the phenomena we aim to explore.
>
> * **Quality of content**: Following this, we assess the quality of the example in terms of language, coherence, and sentence flow. This includes verifying that the output generated from these prompts is accurate and pertinent.
>
> * **Consistency with other data**: To ensure that the chosen examples are not outliers or anomalous data points, they are compared with other data in the study.
>
> * **Instructional value**: Finally, the instructional value of each example is appraised to confirm that it aids in explaining or demonstrating the phenomena under investigation.
>
> Through this multi-step process, we strive to minimize any sensitivity of the LLM to the choice of examples and ensure the robustness of our findings. Sorry for not adding these details to the paper. We already included it in our updated draft.
>
> 2. **How sensitive is the LLM to the choice of these examples?**
>
> We conduct new experiments by designing new prompt examples. Similar to the human evaluation part, we only collect 100 clean sentences on Fr->En with the new prompt examples, keeping the other parts of Figure 1 the same for the ablation study. After collection, we use the same prompt in Appendix A to ask GPT-4 to give its preference, since it shares a similar preference as human annotators.
>
> First, we design two new sets of prompt examples for the bilingual cleaning method to check whether new prompt examples cause large variances for the same cleaning method. The results are:
> | | Left is better | Tie | Right is better |  |
> | --- | :---: |:---: |:---: |--- |
> | Original prompt examples | 18% | 60% | 12% | New prompt examples v1 |
> | Original prompt examples | 20% | 53% | 27% | New prompt examples v2 |
>
> We do see the variance with different prompt examples. However, the major preference (>50%) of GPT-4 is located in "Tie", indicating that the same cleaning method with different prompt examples offers similar cleaned sentences.
>
> Second, we design two more sets of prompt examples for the monolingual and translation cleaning methods to check whether we have a similar pattern as the previous human/GPT-4 evaluation. The results are:
> | | Left is better | Tie | Right is better |  |
> | --- | :---: |:---: |:---: |--- |
> | Original bilingual | **36%** | 40% | 24% | Monolingual with new prompt examples |
> | Original bilingual | **34%** | 35% | 31% | Translation with new prompt examples |
> | Monolingual with new prompt examples| 29% | 31% | **40%** | Translation with new prompt examples |
>
> A similar pattern, i.e. bilingual > translation > monolingual, is observed. **In sum, different prompt examples will cause a mild variance, but the observed pattern is similar.**
>
> ---
>
> > Presentation Improvement: Table 3 numbers are a bit too small to read comfortably.
>
> Thanks for this advice. We already adjusted the font size of Table 3 in our updated draft.
>
> ---
> Appendix A. Prompt for GPT-4 evaluation
> ```
> def generate_prompt(source, target, v1, v2):
>     return """In the following, I'm going to show you one noisy source sentence in French and one noisy target sentence in English. In addition, I also offer you two clean versions of the noisy target sentence.
>
> Can you rank these two clean target sentences based on these three criteria:
> 1. Comprehensive noise removal: All form of noise should be eliminated from the noisy target sentence, including removing semantically meaningless emojis, translating emojis with semantic content into words, correcting misspellings, etc.
> 2. Semantic preservation: The clean target sentence should retain similar semantic information as the original noisy target sentence.
> 3. Alignment with the source sentence: The clean target sentence should convey the same intended meaning as the noisy source sentence, ensuring accurate translation and faithful representation of the original content.
>
> Noisy source sentence in French: {0}
> Noisy target sentence in English: {1}
> The first clean target sentence: {2}
> The second clean target sentence: {3}
>
> For your output, you don't need to give any explanation. If the first version is better, you output 1. If the second version is better, you output 2. If they are equally clean, you output 3.""".format(source, target, v1, v2)
> ```

---

### Meta-Review · Area_Chair_nEJP · 2023-09-18

**Recommendation:** 3

**Metareview:**

This paper presents a method using an LLM (GPT) to automatically remove “noise” from the target side of noisy parallel data (the MTNT dataset, en-fr and en-ja), to make the dataset more useful for testing the robustness of MT models. They compare the performance on 3 input types: monolingual (just the target sentence), bilingual (provide both source and target) and translation (provide just the source sentence). They focus the evaluation on 4 known types of “noise”: spelling/grammar errors, emojis, slang and profanities and compare again a rule-based correction tool. The best method depends on the noise type (e.g. bilingual working best for emojis/slang and the correction tool is best for spelling/grammar errors). They then test the new corpus C-MTNT on the MT task and show that the bilingual method leads to the greatest gain in performance when training MT models to produce clean target sentences from noisy source data.

The paper’s motivation was well appreciated by the reviewers and the experiments are interesting and useful for future research. The authors’ extensive rebuttal provided additional experimental details and justification some design choices, answering most of the reviewers’ comments. The paper was initially lacking some human evaluation which was provided during the rebuttal. I recommend including all this in the paper.

---

### Decision · Program_Chairs · 2023-10-07

**Decision:**

Accept-Findings

**Comment:**

This paper presents a method using an LLM (GPT) to automatically remove “noise” from the target side of noisy parallel data (the MTNT dataset, en-fr and en-ja), to make the dataset more useful for testing the robustness of MT models. They compare the performance on 3 input types: monolingual (just the target sentence), bilingual (provide both source and target) and translation (provide just the source sentence). They focus the evaluation on 4 known types of “noise”: spelling/grammar errors, emojis, slang and profanities and compare again a rule-based correction tool. The best method depends on the noise type (e.g. bilingual working best for emojis/slang and the correction tool is best for spelling/grammar errors). They then test the new corpus C-MTNT on the MT task and show that the bilingual method leads to the greatest gain in performance when training MT models to produce clean target sentences from noisy source data.

The paper’s motivation was well appreciated by the reviewers and the experiments are interesting and useful for future research. The authors’ extensive rebuttal provided additional experimental details and justification some design choices, answering most of the reviewers’ comments. The paper was initially lacking some human evaluation which was provided during the rebuttal. I recommend including all this in the paper.